# A tropospheric pathway of the stratospheric quasi-biennial oscillation (QBO) impact on the boreal winter polar vortex

Koji Yamazaki[1], Tetsu Nakamura[1], Jinro Ukita[2], and Kazuhira Hoshi[3]

[1]Faculty of Environmental Earth Science, Hokkaido University, Sapporo, 060-0810, Japan
[2]Faculty of Science, Niigata University, Niigata, 950-2181, Japan
[3]Graduate School of Science and Technology, Niigata University, Niigata, 950-2181, Japan

*Correspondence to*: Koji Yamazaki (yamazaki@ees.hokudai.ac.jp)

**Abstract.** The quasi-biennial oscillation (QBO) is quasi-periodic oscillation of the tropical zonal wind in the stratosphere. When the tropical lower stratospheric wind is easterly (westerly), the winter Northern Hemisphere (NH) stratospheric polar vortex tends to be weak (strong). This relation is known as Holton-Tan relationship. Several mechanisms for this relationship have been proposed, especially linking the tropics with high-latitudes through stratospheric pathway. Although QBO impacts on the troposphere have been extensively discussed, a tropospheric pathway of the Holton-Tan relationship has not been explored previously. We here propose a tropospheric pathway of the QBO impact, which may partly account for the Holton-Tan relationship in early winter, especially in the November-December period. The study is based on analyses on observational data and results from a simple linear model and atmospheric general circulation model (AGCM) simulations. The mechanism is summarized as follows: the easterly phase of the QBO is accompanied with colder temperature in the tropical tropopause layer, which enhances convective activity over the tropical western Pacific and suppresses over the Indian Ocean, thus enhancing the Walker circulation. This convection anomaly generates Rossby wave train, propagating into the mid-latitude troposphere, which constructively interferences with the climatological stationary waves, especially in wavenumber 1, resulting in enhanced upward propagation of the planetary wave and a weakened polar vortex.

## 1 Introduction

The stratospheric quasi-biennial oscillation (QBO) is dominant interannual oscillation of the zonal wind in the stratospheric tropics with an approximate 28-month period (Veryard and Ebdon, 1961; Reed et al., 1961; Baldwin et al., 2001). The influence of QBO on the winter Northern Hemisphere (NH) stratospheric polar vortex has been well known (Holton and Tan, 1980, 1982; Anstey and Shepherd, 2014). When the tropical lower stratospheric wind is easterly (EQBO) the winter NH polar vortex tends to be weak, and the vortex tends to be strong when the tropical lower stratospheric wind is westerly (WQBO). This relation is called the Holton-Tan relationship (Holton and Tan, 1980, 1982), for which several mechanisms have been proposed in terms of the stratospheric linkages between the tropics and high-latitudes (Anstey and Shepherd, 2014). In the EQBO winters, the westerly region in the lower stratosphere is limited poleward of around 20°N so that the waveguide for quasi-steady planetary waves becomes narrower. Thus the planetary waves tend to propagate more poleward and weaken the polar vortex. On the other hand, in the WQBO winters, the westerly region extends more to the tropics, and thus planetary waves tend to propagate more equatorward. Holton and Tan (1980, 1982) only showed a plausible mechanism, as the latitudinal position of the zero-wind critical surface of stationary Rossby wave is primarily controlled by the equatorial QBO. Recently, Watson and Gray (2014) posted this line of discussion with their model. Naoe and Shibata (2010) analyzed Holton-Tan relationship by a QBO-producing chemistry-climate model (CCM) and reanalysis data. They showed the conventional critical latitude mechanism that the equatorial winds in the lower stratosphere acted as a waveguide for

planetary wave propagation did not hold. White et al. (2015) suggested the enhanced upward wave propagation at mid-latitudes due to the enhanced wave growth rather than the critical latitude mechanism, explaining the QBO-related change in mid-latitudes as well as the polar vortex change in high-latitudes. Naoe and Shibata (2010) and Yamashita et al. (2011) suggested the importance of the secondary circulation induced by the equatorial QBO in the middle stratosphere rather than the lower stratosphere. Garfinkel et al. (2012) and Lu et al. (2014) pointed the significance of the QBO-induced meridional circulation anomalies extending from the subtropics to mid-latitudes through changes in the refraction index and modulation of Rossby wave propagation.

The Holton-Tan relation has been the subject of many observational and modelling studies, yet its underlying mechanism may not be so completely understood (Anstey and Shepherd, 2014). The mechanisms mentioned above are processes linking the equatorial stratosphere to the polar stratosphere through the stratosphere, thus referred to as the stratospheric pathway in this study.

The influence of the QBO on the troposphere has been also the subject of many studies (Baldwin, 2001; Marshall and Scaife, 2009; Gray et al., 2018). In the EQBO winters, planetary wave in the troposphere especially of wavenumber 1 is enhanced compared with the WQBO years in mid- to high-latitudes (Baldwin and Dunkerton, 1991; Hu and Tung, 2002, Ruzmaikin et al., 2005; Naoe and Shibata, 2010). This has been interpreted as a stratospheric influence on the troposphere, by changing stratospheric zonal wind distribution from the tropics to high-latitudes, then changing propagation property of the stratosphere. Previous studies have reported that the tropical convection is also affected by the QBO phase (Collimore et al., 2003; Liess and Geller, 2012; Gray et al., 2018). Particularly, the impact of the QBO on the Madden-Julian oscillation (MJO) (Madden and Julian, 1994) has been extensively examined in recent years (Yoo and Son, 2016; Son et al., 2017; Marshall et al., 2017; Nishimoto and Yoden, 2017; Martin et al., 2019; Hood et al., 2020). In the EQBO winters the MJO is more active compared with the WQBO winters. By using a local cloud-resolving WRF model, Martin et al (2019) showed that the colder temperature anomaly in the tropical tropopause layer (Fueglistaler et al., 2009) associated with the EQBO phase is an essential factor for enhancing tropical deep convection. Those results potentially suggest that resultant changes in the tropical convection from the QBO may also influence high-latitude circulations, namely the extra-tropical planetary-scale wave field and the stratospheric polar vortex strength, through tropospheric processes. Peña-Ortiz et al (2019) examined QBO influence the tropical convection and showed QBO modulation of the tropical convection that impacts stationary waves and the polar vortex of the austral winter of the southern hemisphere. However, such a tropospheric pathway for the Holton-Tan relation in the Northern Hemisphere has not been studied.

In this study, we present evidence for a possible mechanism of the tropospheric pathway for the Holton-Tan relationship through the following process. 1) The QBO affects tropical convection. 2) The tropical convection then affects mid-latitude planetary waves. 3) Finally the upward planetary waves propagation to the stratosphere is modified. Although there have been many studies discussing on each of those processes, our aim is to provide a synthetic view on potential QBO influences through a tropospheric pathway by analyzing observations and results from a simple linear model and AGCM simulations. It should be noted that we intend to argue for a tropospheric process for the mechanism of Holton-Tan relation and QBO influence on NH weather in early winter, but not to deny a role of the stratospheric pathway.

## 2 Data and methods

### 2.1 Data and analyses

We used 6-hourly and monthly-mean atmospheric variables from the ERA-interim reanalysis data (Dee et al., 2011), at a 1.5° horizontal resolution and 37 vertical levels (1000–1 hPa). We also used monthly-mean outgoing long-wave radiation (OLR) data from the National Oceanic and Atmospheric Administration (NOAA) interpolated OLR site (https://www.esrl.noaa.gov/psd/data/gridded/data.interp_OLR.html), at a 2.5° horizontal resolution. Both datasets were analysed from 1979/80 to 2015/16 (37 boreal winter seasons). The Eliassen-Palm (EP) flux (Andrews and McIntyre, 1976) values were calculated from the 6-hourly data. The sunspot number data have been obtained from the World Center for Sunspot Index and Long-term Solar Observation (WDC-SILSO), Royal Observatory of Belgium, Brussels (http://www.sidc.be/silso/datafiles; Clette et al. 2014).

### 2.2 Definition of the phase of the QBO

The phase of the QBO was defined using the winter (DJF) averaged zonal-mean zonal wind at 50 hPa averaged over 5°S to 5°N. The winters were classified as WQBO or EQBO winters when the absolute values exceeded 3 m s$^{-1}$. If the direction of the equatorial 50 hPa zonal wind changed during winter, we excluded that winter. These criteria resulted in 19 WQBO and 12 EQBO winters, respectively. The analysis was based on composite analysis for EQBO and WQBO winters. Recognizing high frequencies of La Niña and El Niño events defined by the Japan Meteorological Agency (JMA) in the EQBO and WQBO winters, respectively, we also made the composite analysis in which ENSO (El Niño and La Niña) winters were excluded (see Table 1). The definition of ENSO used by the JMA is based on 5-month moving averaged SST deviation from the standard value at NINO.3 (5°S-5°N, 150°W-90°W). When the SST deviation experiences more (less) than +0.5K(-0.5K) over 6 consecutive months, it is defined as El Niño (La Niña). The standard value is defined by previous 30-year mean for each month. Without ENSO winters, we have 9 WQBO and 7 EQBO winters (see Table 1). In the following, the analyses with and without ENSO winters are shown and discussed. We also examined two cases in which we changed the threshold wind speed set to 0 m/s (14 EQBO and 23 WQBO winters) and the reference height to 40 hPa (18 EQBO and 14 WQBO winters). In both cases, the results show a high degree of robustness.

### 2.3 Linear model experiments

Atmospheric response to a prescribed diabatic heating was calculated by a linear baroclinic model (LBM) (Watanabe and Kimoto, 1999, 2000) for a given climatological basic state and a thermal forcing. The LBM is a diagnostic tool used to simulate a linear tropospheric response to an anomalous forcing (e.g., Otomi et al., 2013). We used a spectral resolution of T42 with 20 vertical layers. In an experiment, the vertical maximum of the heating is placed at 500 hPa with a maximum heating rate of 1 K/day, which is comparable with an actual QBO-signal. Here, the QBO-signal refers to the EQBO minus WQBO difference rather than the deviation from climatology. The extent of the horizontal heating domain is 40° in longitude and 12° in latitude, and the magnitude of heating linearly decreases to the domain boundary (see Appendix A in detail). The basic state is based on the monthly climatology (averaged 1979–2010) from the National Centers for Environmental Prediction–National Center for Atmospheric Research (NCEP–NCAR) reanalysis (Kalnay et al., 1996). Note that the linear response by the LBM is only meaningful in the middle troposphere to the lower stratosphere, because temperature and wind of the response in near-surface levels and above the middle stratosphere are strongly damped to zero with a time scale of about 1 day.

### 2.4 AGCM experiments

An AGCM used in this study is the AGCM for the Earth Simulator (AFES) (Ohfuchi et al., 2004) version 4.1 with triangular truncation at horizontal wavenumber 79, and with 56 levels and the top level of about 0.1 hPa (T79L56). The AFES version 4.1 was used for studying impacts of sea ice on mid-latitude climate and its stratospheric role (Nakamura et al., 2015, 2016). As the boundary condition, we used monthly mean data from the Merged Hadley–NOAA/Optimum interpolation (OI) sea surface temperature (SST) dataset (Hurrell et al., 2008). The 30-year average of 1981-2010 was used as the prescribed SST for three types of simulations. One is a "Control" experiment, in which the AFES model was integrated for 60 years with the climatological boundary conditions. The second simulation is a "CONV1" experiment, in which anomalous convective heating was placed over the western tropical Pacific centered at 150°E, 5°N. The third simulation is a "CONV2", in which anomalous convective cooling was placed over the Indian Ocean centered at 70°E, 5°N, together with the western Pacific anomalous heating of CONV1. The CONV1 and CONV2 have 60 ensembles of a 1-year integration branched from 1 July in each year of the Control experiment.

### 2.5 Analysis method and statistical significance

For observational analyses for OLR and other meteorological fields, we used composite analyses based on the QBO phase. The statistical significance was calculated using the two-sided Student's $t$ test for the composite difference. For AGCM experiments, the difference between CONV1 (CONV2) and Control experiment averaged over 60 years was analysed. The statistical significance was calculated using the two-sided Student's $t$ test for the difference.

### 3 Results

### 3.1 Reconfirmation of the Holton-Tan relationship

Fig.1 confirms the extratropical QBO signal (EQBO minus WQBO) in composite differences in zonal wind and temperature fields, which is robust even if we exclude ENSO winters (Fig. 2). The timeseries of extratropical QBO signal for the zonal mean zonal wind ([U]) at 60°N indicates deceleration in mid-November till late January for both all and without ENSO winters (Fig. 1a and Fig. 2a). Three-month (November-January) mean difference of [U] shows statistically significant negative signals at the mid- to high-latitude in the stratosphere for both composite cases (Fig. 1b and Fig. 2b), and extending into the troposphere around 60°N for the without-ENSO case (Fig. 2b). The maximum value of [U] signal reaches -10 m/s at around 10 hPa, 65°N for both cases. The zonal mean temperature ([T]) shows warm Arctic signals in thermal valance with [U] signal (Fig. 1c and Fig. 2c). Notably, the tropical temperature in the layer from 50 to 120-hPa shows a statistically significant cold signal, which is balanced with negative vertical [U] shear (Plumb and Bell, 1982).

EP flux difference shows the poleward QBO signals from the equator to mid-latitudes in the upper troposphere (arrows in Fig. 1b and Fig. 2b). The upward QBO signals from the mid-latitude troposphere into the high-latitude stratosphere are also seen. Although previous studies (e.g. Naoe and Shibata, 2010; Yamashita et al., 2011) have noticed this feature, they focused more on mid-winter characteristics of circulation anomalies. Instead, we shall examine the tropical convective activity in early winter by recognizing the tropics as an origin of this QBO signal in the poleward and upward flux.

### 3.2 QBO signal on the tropical convection and circulation

The OLR difference of the QBO signal (EQBO minus WQBO) in early winter (OND) shows prominent negative signals over the tropical western Pacific and the South Pacific Convergence Zone (SPCZ), denoting enhanced convective activity there (Fig. 3a and Fig. 4a). The OLR signal also shows suppressed convective activity over the Indian Ocean. Those QBO-

related convective features do not change even if ENSO winters are excluded (Fig. 3a and 4a), although the suppressed convection over the central to eastern equatorial Pacific appears for the ENSO-included case due to more La Nina cases in
the EQBO composite (Table 1).

The relationship between the QBO and tropical deep convection has been studied in previous studies (Collimore et al., 2003; Liess and Geller, 2012; Gray et al., 2018; Martin et al., 2019). For example, Collimore et al. (2003) studied the relationship between the QBO and tropical convection by analyzing observations of highly reflective cloud and the OLR, obtaining similar results to ours. Martin et al.(2019) examined the impact of the QBO on the local convection using a
regional cloud-resolving model, and found that the cold temperature near the cloud top enhances the tropical convection. From those previous works and present analyses, we suggest the following scenario. At the tropical tropopause (around 100 hPa), the western Pacific is climatologically the coldest region in the tropics (Fig. 3b and 4b). SST being the highest over this region, and near tropopause-level temperature anomalies in the EQBO winters (Fig. 3c, 4c) likely provide favorable conditions for enhanced convective activity. Although we do not know precise mechanisms by which negative tropopause
temperature anomalies provide favorable conditions for enhanced convective activity, we suspect weak stability and subsequent increase in cloudiness in the tropical tropopause layer (TTL) are the main two key elements. In addition feedback arising from cooling in the TTL and warming in the mid-troposphere by cloud longwave forcing may farther accelerate weak stability, thereby enhancing the convective activity, as noted by Giorgetta et al. (1999) for boreal summer season.

The enhanced convection over the tropical western Pacific is accompanied by suppressed convection over the western
Indian Ocean, where the downward branch of Walker circulation lies. Indeed, the longitude-height section of circulation anomalies (EQBO-WQBO) at the equatorial belt (10°S-10°N) clearly indicates the enhancement of the Walker circulation with an upward branch over the tropical western Pacific (120°E-170°E) and a compensating downward branch over the Indian Ocean (40°-90°E) in early winter (Fig. 5).

Next, we compare diabatic heating between the EQBO and WQBO winters over the tropical western Pacific (130°E-
160°E, 0°-10°N) from the conservation law of potential temperature using the ERA-Interim data (Fig. 6). Detailed calculation method is shown in Appendix B. Diabatic heating in the EQBO years is larger than that in the WQBO years during the October-December period with or without ENSO years. The difference becomes most prominent in the middle troposphere in November. The maximum difference is about 1 K/day and is statistically significant in the case of with ENSO years, and nearly significant without ENSO years.


**3.3 QBO signal on the extra-tropical circulation in November**

Hereafter we focus on November because the tropospheric pathway is most clearly seen in November, which will be shown in Fig.16. The QBO signal on the geopotential height at 250 hPa (Z250) is shown in Fig. 7. At the upper troposphere, the troughs over Siberia and over the North Pacific are seen both with and without ENSO years. In particular, a trough over
Siberia is significant (Fig. 7a and 7c). The stratospheric polar vortex is weakened in EQBO Novembers, though the signal is not significant in without ENSO Novembers.

When the QBO signal (EQBO minus WQBO) in Z250 is decomposed into its wavenumber components (Fig. 8a and 8c), a pair of negative and positive anomalies appear over climatological trough and ridge regions in the mid-latitude (around 50°N) for the wavenumber 1, implying intensified upper tropospheric planetary waves for EQBO. For wavenumber 2,
interference shifts wavenumber 2 field eastward and slightly enhances the amplitude (Fig. 8b and 8d). Fig. 9 shows wave amplitudes at 250 hPa as a function of latitude in the different QBO phases. Red lines are based on EQBO composites and blue lines from WQBO composites. Peak values of wave amplitude increase in EQBO Novembers both for wave-1 and wave-2 and regardless of all or non-ENSO composites. The linear interference between the Rossby wave response and background climatological stationary wave has been studied in previous studies, e.g. the interference between extratropical

surface forcing and the annular mode (Smith et al., 2010), the tropospheric precursor and the stratospheric polar vortex (Garfinkel et al., 2010), and the solar maximum and westerly QBO (Yamashita et al., 2015).

**3.4 Response of NH mid-latitude to the tropical convection –LBM experiments**

We next used the LBM to attempt to link the above mentioned circulation signal to the tropical convective heating signal. The heating distribution is shown in Appendix A. The simulated linear response of the geopotential height at 250 hPa (Z250) to an anomalous heating placed over the western tropical Pacific (150°E, 5°N) under the November climatological condition is characterized by a decreased height over the northwest Pacific, and strengthened subtropical jet around 35°N and weakened polar jet in the troposphere and the lower stratosphere (Fig. 10a and 10b). The simulated Z250 linear response to an anomalous cooling placed over the western Indian Ocean (70°E, 5°N) shows a wavy pattern over the Pacific and positive anomalies over Canada and western Russia (Fig. 10c). The both subtropical and polar night jets are weakened (Fig.. 10d). The combined response of the above pair is a pronounced wave pattern over the Pacific-North America sector with negative response extending into east Siberia (Fig. 10e). The high-latitude jet in the troposphere and the lower stratosphere are reduced (Fig. 10f) similar to the observed QBO signal (Fig. 7).

Fig.11 illustrates the negative anomalies centred over the northwest Pacific from the wave number 1 component of the linear response, which constructively interfere with the climatological trough (Fig.11a and 11c). For wavenumber 2, the linear responses are shifted eastward against the climatological field (Fig.11b and 11d). This linear response deepens the Pacific trough and enhances the Atlantic ridge, thus providing a favourable tropospheric condition for the stratospheric polar vortex weakening (e.g., Garfinkel et al., 2010), especially for the stratospheric sudden warming later propagating to the troposphere (Nakagawa and Yamazaki, 2006). In summary, the anomalous convective heating over the tropical western Pacific generates the wavenumber 1 anomaly in the NH extratropics, which is constructive to the climatological wavenumber 1 field.

Where is a preferable location of convection to interfere constructively with the NH climatological eddy field? To address this question we placed a convective heating (see Section 2.3 in details) at a 20° interval in longitude, and a 15° interval in latitude, and calculated spatial correlations between the linear responses and the climatological eddy fields poleward of 40°N. This correlation map was made for wavenumbers 1 and 2 and from October to December, separately. At 250 hPa (Fig. 12), convection over the Pacific region gives rise to strong constructive interference with the NH climatological eddy field. In particular, the western tropical Pacific is the most preferable location for the constructive interference, especially for November and wavenumber 1. For wavenumber 2, convection over the western tropical Pacific also works as a constructive player from October to December.

The above LBM-based linear analysis provides qualitatively consistent results with the observed QBO signal. However, the response is one order of magnitude weaker than the observed QBO signal. This is probably because the LBM calculation includes no interaction between the anomalous response and climatological fields, no stratosphere-troposphere coupling, nor feedbacks from transient eddies, since the LBM has the interference for linear processes only. To attempt to further elucidate the role of the tropical convection induced by the QBO, we performed the AGCM simulations.

**3.5 Response of NH mid-latitude to the tropical convection –AGCM experiments**

Two AGCM experiments were made in addition to a control experiment. One is the response to anomalous tropospheric heating over the western tropical Pacific (CONV1), and the other is the response to the pair of heating over the western tropical Pacific and cooling over the Indian Ocean (CONV2). These heating anomalies are related to the QBO signals in the OLR both for all and without-ENSO cases (Fig. 3a and 4a).

The NH extratropical response is shown in Fig. 13, which can be compared with Fig. 7. The November response of Z250 by the AGCM is similar to that by the LBM (Fig. 10) but with increased magnitude. The same as in the LBM case, the eddy response of Z250 is similar to the eddy climatology from the AGCM control run (Fig. 14), implying constructive interference of the response from the tropical convection with the climatological eddy field. Wave amplitudes at 250 hPa for all simulations are shown in Fig. 15. Compared with the observed QBO difference (seen as EQBO minus WQBO in Fig. 9), simulated differences between CONV1 (CONV2) and CNTL are similar in magnitude and latitudinal profile. For example, simulated wave-1 amplitude averaged over 30-60°N is 12.0 m (CONV1 minus CNTL) and 10.7 m (CONV2 minus CNTL), which is in good agreement with the observed difference of 14.3 m between EQBO and WQBO. Wave-1 amplitude is also peaked at around 55°N for all cases. We also confirm that convection over the tropical western Pacific is most significant for enhanced extratropical planetary wave.

The zonal-mean zonal wind ([U]) is reduced by about 2.5-3 m/s, at 10 hPa, 70°N (Fig. 13b and d), which is similar magnitude as an observed QBO signal (Fig. 7b and d). The subtropical region, on the other hand, shows the large difference between AGCM and observed signals. This is because in the AGCM simulation QBO is not directly represented and thus the stratospheric meridional circulation anomalies in the subtropics are not well represented.

We think that in the present context of the QBO impacts the resemblance between composite differences (EQBO minus WQBO) with and without ENSO (both El Nino and La Nina) events mostly ruled out possible compound influences in mid- to high-latitudes from ENSO (please see Figure 7). However, some differences in the Walker circulation between two composite differences with and without ENSO events, especially in sinking branches (Figure 5a and b). Noting this we have made a series of AGCM experiments. In addition to CONV1 (heating in the western tropical Pacific) and CONV2 (heating in the western tropical Pacific and cooling in the tropical Indian Ocean), results from the experiments with adding negative convective heating placed in the central tropical Pacific around 150°W, 0°N (CONV3P) and in the tropical Atlantic around 30°W, 10°N (CONV3A) are analyzed. In fact, the setting for CONV3A with two sinking branches, one in the Indian Ocean and the other in the Atlantic Ocean mimics the QBO signal without ENSO most (Fig. 5b). The mid- to high-latitudes horizontal pattern in geopotential height anomalies at 250 hPa and zonal-mean zonal wind anomalies (not shown) are similar to the observed QBO signal (Fig. 7b). But most significantly, those horizontal and meridional patterns are captured in all experiments including CONV1 with heating only in the western tropical Pacific. We interpret this that the western tropical Pacific is the most influential to extra-tropics and polar vortex.

**3.6 Seasonal march**

Fig. 16 provides information on the seasonal march of anomalies in the stratospheric polar vortex strength and the upward EP flux at 100 hPa. Blue lines are based on reanalysis data and red lines from the simulated results. The observed QBO signal of anomalous upward EP flux at 100 hPa averaged over 35°-70°N reaches its maximum in November or December depending on if ENSO winters are included (Fig. 16a). The simulated anomalous upward EP flux also reaches its maximum value in November for CONV1 and December for CONV2, though the magnitude is about a half of the observed value. The observed QBO signal in the stratospheric zonal-mean zonal wind (55°-80°N, 50 hPa) is larger in December and January, and the signal becomes small in February (Fig. 15b). The effect of the tropospheric pathway estimated by the AGCM shows the maximum in November and weakens in December. The effect of the tropospheric pathway is comparable with the observed QBO signal of the polar vortex strength in November, and it is small in December.

Why the tropospheric mechanism does not seem to occur from January onward? Is this because QBO-induced tropical convection anomalies disappear in midwinter? We examined the observed tropical convection difference between EQBO and WQBO for each month and found that it does not disappear but shifts slowly eastward. We then made simple diagnostics on seasonal change in observed wave amplitudes (Fig. 17). At 250 hPa, from September to November, wave-1

amplitude in EQBO is larger than that in WQBO at around the maximum latitude. This means that the maximum wave-1 amplitude is enhanced. In December, wave-1 amplitude in EQBO is enhanced in high-latitudes. Although this high-latitude enhancement continues to March, wave-1 amplitude at the maximum latitude of 50°N is reduced and no wave-1 amplitude enhancement in the troposphere is seen from January onward. On the contrary, the stratospheric polar vortex in EQBO weakens more in January (Figs. 1a and 16b). This corresponds to an enhancement of wave-1 amplitude at 100 hPa (Fig. 17f).

Apparently, wave-1 amplitude in EQBO becomes larger than that in WQBO from November to February. For wave-2, the seasonal march at 100 hPa and that at 250 hPa are similar (not shown). We suppose the stratospheric processes discussed in many previous studies can account for the mid-winter Holton-Tan relationship. In mid-winter, the stratosphere undergoes vacillation without changes in the troposphere (Holton and Mass, 1976; Chen et al., 2001; de la Cámara et al., 2019).

**3.7 Modulation by 11-year solar cycle**

It has been known that the Holton-Tan relation is modified by the 11-year solar cycle (Labitzke, 2005, and references therein). Recently, Misios et al. (2019) provided strong evidence for weakened Walker circulation at the solar maximum. Recognizing possible compounding influences by the solar cycle on the QBO impact on tropical convection and extra-

tropical circulation anomalies as discussed in our paper so far, we have made additional composite analysis as follows.

We used the Nov-Feb mean sunspot number as a solar index (SSN; Fig. A2) whose average value is 92.2. Winters above (SSN>92.2) and below (SSN<92.2) the average are classified as solar max and solar min winters, respectively. We also divided winters into EQBO, WQBO composites, and other winters as described in Section 2.2 (see Table A1 for the sample size of each category). As identified in Misios et al. (2019), the solar impacts on convective activity thus the Walker

circulation have one to two years of time lag through the bottom-up mechanism. We thus shifted by one year when classifying solar max and min winters. This sampling scheme provides consistent results with theirs on the solar influence, i.e. stronger Walker circulation at solar minimum seen in Figure A3c.

The QBO signal (EQBO minus WQBO) in OLR is stronger in the solar min years with significantly enhanced convection over the western tropical Pacific. In the solar max years, enhanced convection over the western tropical Pacific is weaker and

shifts eastward slightly. Despite some differences, the QBO signal characterized by enhanced convection in the western tropical Pacific is commonly found in both solar max and min composites.

**4 Discussion and conclusions**

From the composite analysis of the observed data we found that convective activity over the tropical western Pacific is enhanced and that over the Indian Ocean is suppressed in the EQBO compared with WQBO winters. We also made a regression analyses of OLR on the polar night jet, which also shows the robust relationship between the polar night jet and tropical convection, especially over the western tropical Pacific and in early winter.

The linear response of the NH atmospheric circulation in the troposphere to the tropical convection anomalies showed

significant constructive interference with the climatological eddy field. Thus the planetary wave, particularly the wavenumber 1, is enhanced due to the tropical convective anomalies. However, this linear response was much smaller than the observed QBO-related signal. The AGCM simulations prescribing a pair of convective heating or at least that over the western tropical Pacific mediated this shortcoming. Based on our analyses we argue that the tropospheric pathway for the Holton-Tan relation plays a major role in November, and a smaller role in December, while the stratospheric pathway plays

a more dominant role from December onward. The tropospheric pathway acts to enhance the NH tropospheric planetary waves, thus influencing the tropospheric circulation. In particular, EQBO tends to deepen the Aleutian low in November and

East Asia experiences cold anomalies. Such anomalies associated with the QBO partly come from the tropospheric pathway through the tropical convection.

One might think that the AGCM can be used to examine the tropospheric pathway by only prescribing the tropical lower stratospheric temperature associated with the QBO. We have tried such an experiment. However, we have recognized two major shortcomings. One is that a parameterized cumulus convection scheme used in the AGCM cannot guarantee a faithful response of convection to associated temperature anomalies in the tropical tropopause layer. The other is that the wind field would change to balance the temperature field, and thus it is very hard to distinguish between the tropospheric pathway and

the stratospheric pathway. Thus we did the experiments specifying the tropical convective heating associated with the QBO. Related to such difficulty, this study has an implication for the model development. Improvement of the representation of cumulus convection especially its behavior in the tropical tropopause layers in the global climate model has benefits to not only representation of the local tropical climate variations but also that of the remote mid- and high-latitude climate variations.


*Data availability.* All observed data used in this study were based on data publicly available, as described in Sections 2.1 and 2.1.

The OLR data is available at: https://www.esrl.noaa.gov/psd/data/gridded/data.interp_OLR.html.

The LBM and AGCM simulation data used in this study are available from the corresponding author upon reasonable

request.

*Code availability.* ALL codes used for analyses of the reanalysis and simulation data are available from the corresponding author upon reasonable request.

**Appendix A: Shape and magnitude of prescribed heating rate**

To examine the atmospheric response to the tropical convection anomaly, we performed the LBM and AGCM simulation using ideal heating that mimics tropical convection. $J$ is determined as

$$Av = \exp[-15 \cdot (\sigma - 0.5)^2]$$

$$h = \left[\frac{(lon - 150)^2}{40^2} + \frac{(lat - 5)^2}{12^2}\right]^{0.5}$$

$$Ah = 1.0 - h, for\ h \leq 1.0$$

$$Ah = 0, for\ h > 1.0$$

$$J = A_{max} \cdot Av \cdot Ah$$

where $Av$ is the vertical distance factor, $Ah$ the horizontal distance factor determined by $h$, and $A_{max}$ the maximum amplitude of heating. Fig. A1 shows the case with the maximum of the heating of 1.0 K d$^{-1}$ located at 0.5 sigma level, 150°E and 5°N, with reduced amplitude linearly with horizontal distance and exponentially with veridical distance from the center.


**Appendix B: Evaluation method of diabatic heating rate**

We evaluated diabatic heating due to anomalous tropical convection based on the conservation law of potential temperature. We assumed that horizontal advection is negligible in the tropics as well as the tendency term, considering the monthly time scale and that its composite mean is close to be in equilibrium. Then, the equation adopted in this study is

$\omega\ \partial\theta\ /\ \partial p = Q = J\ (p_0/p)^{\kappa}$

where $\omega$ is the pressure velocity, $\theta$ the potential temperature, $p$ the pressure, and $Q$ is the diabatic heating in potential temperature, $p_0$ is reference pressure, $\kappa$ is ratio of gas constant to specific heat at constant pressure, and $J$ is the diabatic heating in temperature, for which composite anomalies for the QBOE and QBOW years are shown in Figure 6. Actual calculation is done at each grid of the ERA interim data, using the central differencing method on the pressure coordinate.

*Author contributions.* KY initially got the idea of the tropospheric pathway and designed the study with other co-authors. TN performed AGCM simulations and prepare the LBM calculation. KY analysed the results. The manuscript was prepared with contributions from all co-authors (KY, TN, JU, and KH).

*Competing interests.* The authors declare that they have no conflict of interest.

*Acknowledgements.* AGCM simulations were performed on the Earth Simulator at the Japan Agency for Marine-Earth Science and Technology. The Arctic Challenge for Sustainability (ArCS) program, Belmont Forum InterDec Project, and Grant from WNI WxBunka Foundation supported this study. Thanks to two anonymous reviewers, Elisa Manzini, Lon L. Hood, and the editor, Peter Haynes for providing valuable comments that improved the manuscript.

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

**Table 1** List of EQBO, WQBO, and other years categorized in this paper


| EQBO years | WQBO years | Other years |
| --- | --- | --- |
| 1980, 1982, 1985[La] | 1981, 1983[El], 1986, 1988[El] | 1984, 1987, 1992 |
| 1990, 1997, 1999[La] | 1989[La], 1991, 1994, 1996[La] | 1993, 1995, 2001 |
| 2002, 2004, 2006[La] | 1998[El], 2000[La], 2003[El], 2005 | |
| 2008[La], 2013, 2015[El] | 2007, 2009, 2010[El], 2011[La] | |
| | 2012, 2014, 2016[El] | |


The year denotes that of January. Superscript "El" denotes El Niño winter, and that "La" denotes La Niña winter, both defined by Japan Meteorological Agency.

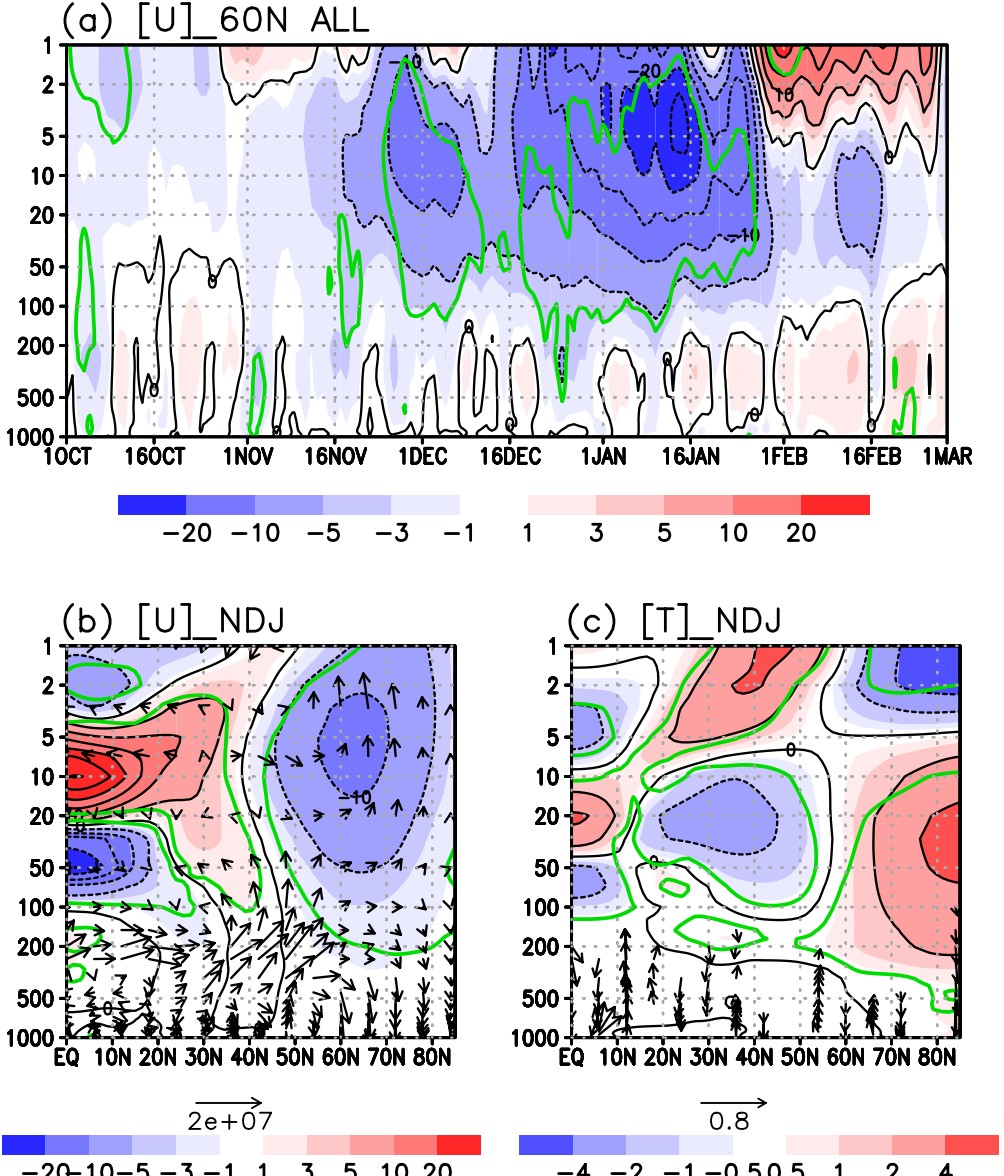

**Figure 1.** (a) Time-height(pressure) section of the composite difference in zonal mean zonal wind at 60°N between EQBO and WQBO winters. Contour interval is 5 m/s. Green line denotes the statistically significant value at 95% confidence. (b) Latitude-height section of the composite difference in zonal mean zonal wind for 3-month (November, December, and January) mean. Contour interval is 5m/s. The arrows are composite difference in EP flux divided by square root of air density. The Unit of EP flux is kg s$^{-2}$ and the scale arrow is shown at the bottom. Vertical component of EP flux is multiplied by 200. (c) The same as (b) but for the zonal mean temperature. Arrows are composite difference in zonal-mean meridional wind (m/s) and reversed zonal-mean vertical p-velocity (ω; hPa/s). p-velocity is multiplied by 200. Values of Δ|ω| less than 5.×10$^{-4}$ Pa/s are omitted.

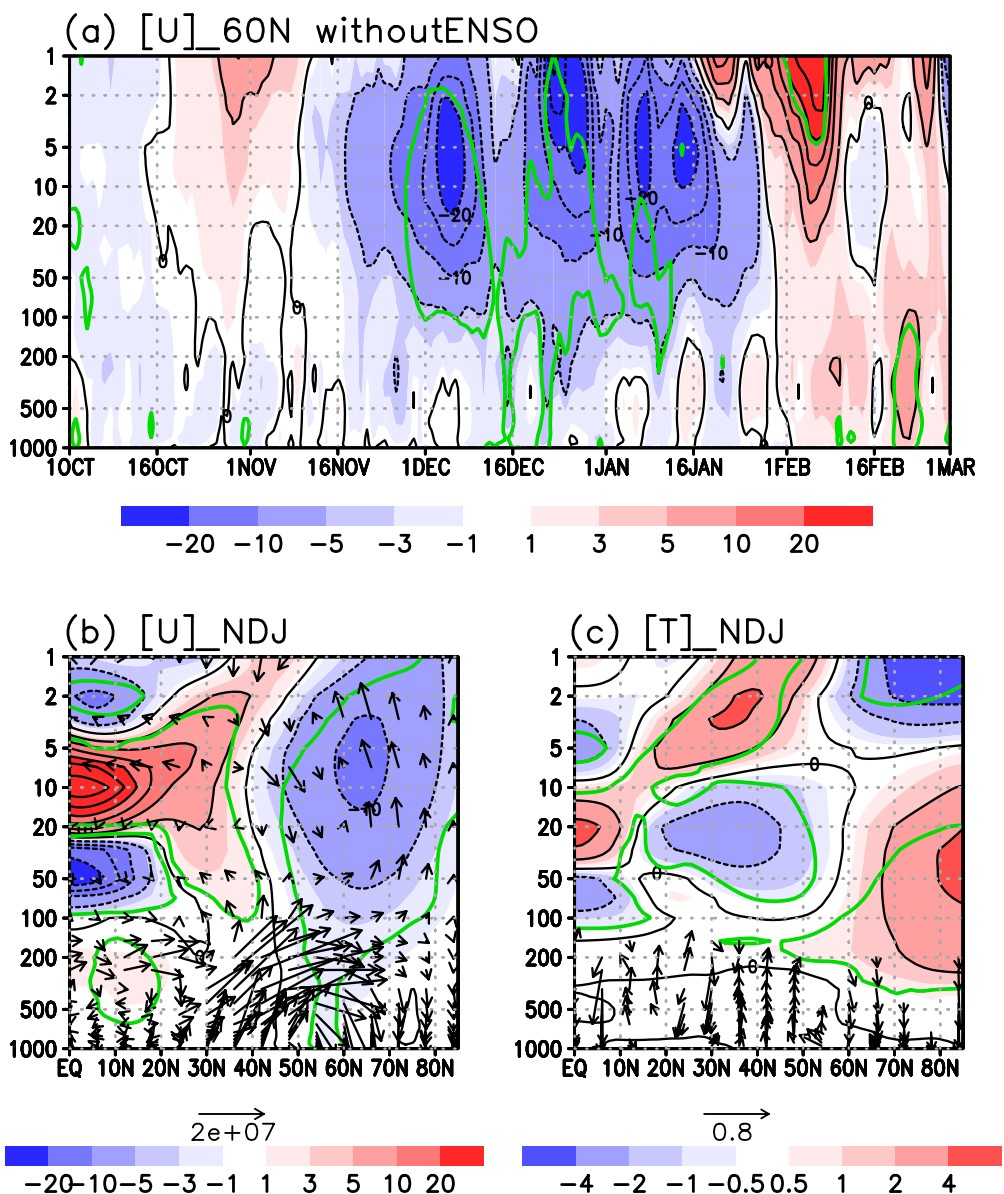

**Figure 2.** Same as Fig. 1 but without ENSO winters.

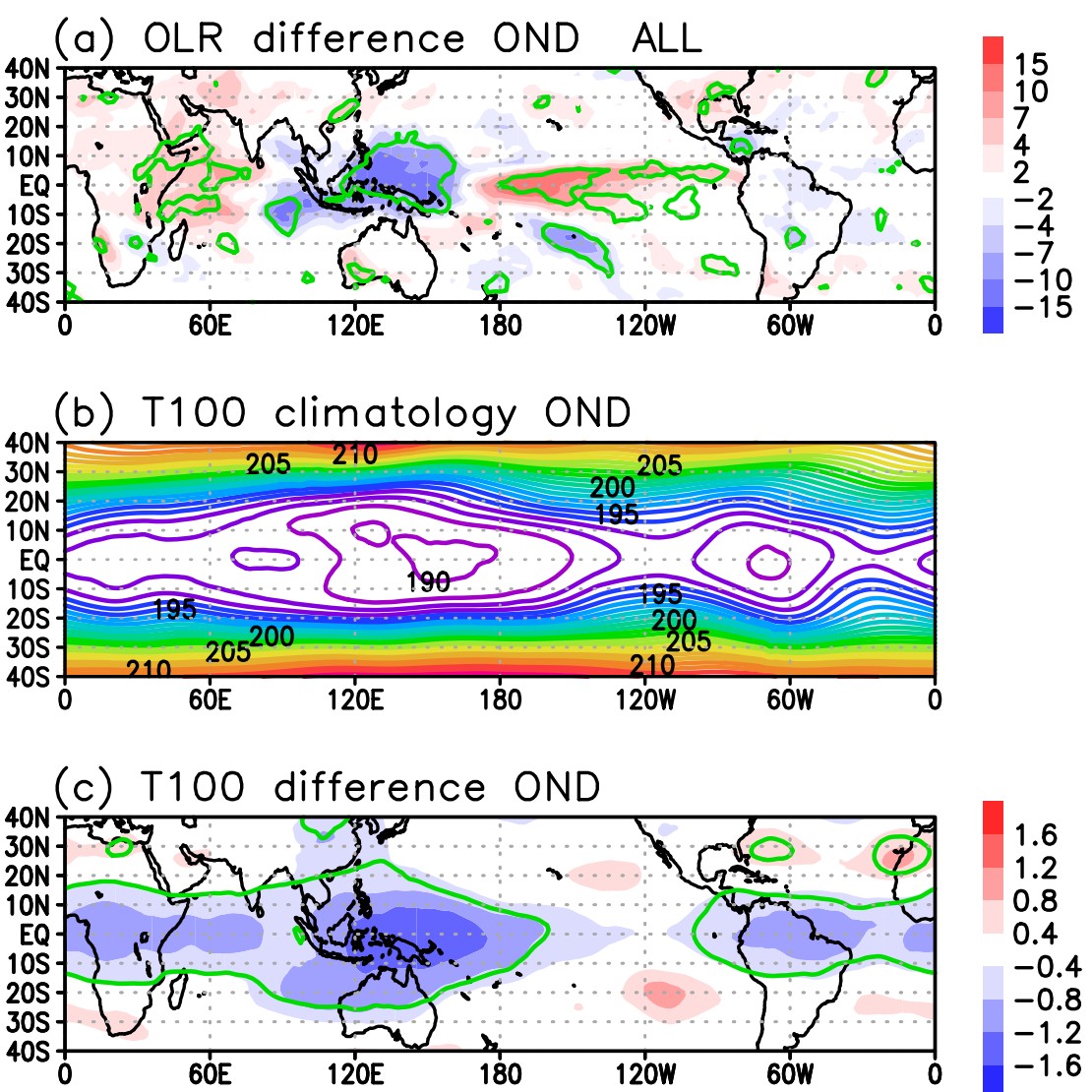

525

**Figure 3.** October-November-December (OND) mean OLR differences between EQBO and WQBO winters. (b) OND-mean temperature at 100 hPa averaged for the composite years. (c) OND-mean temperature difference between EQBO and WQBO winters. Green line in (a) and (c) denotes the statistically significant value at 95% confidence level.

530

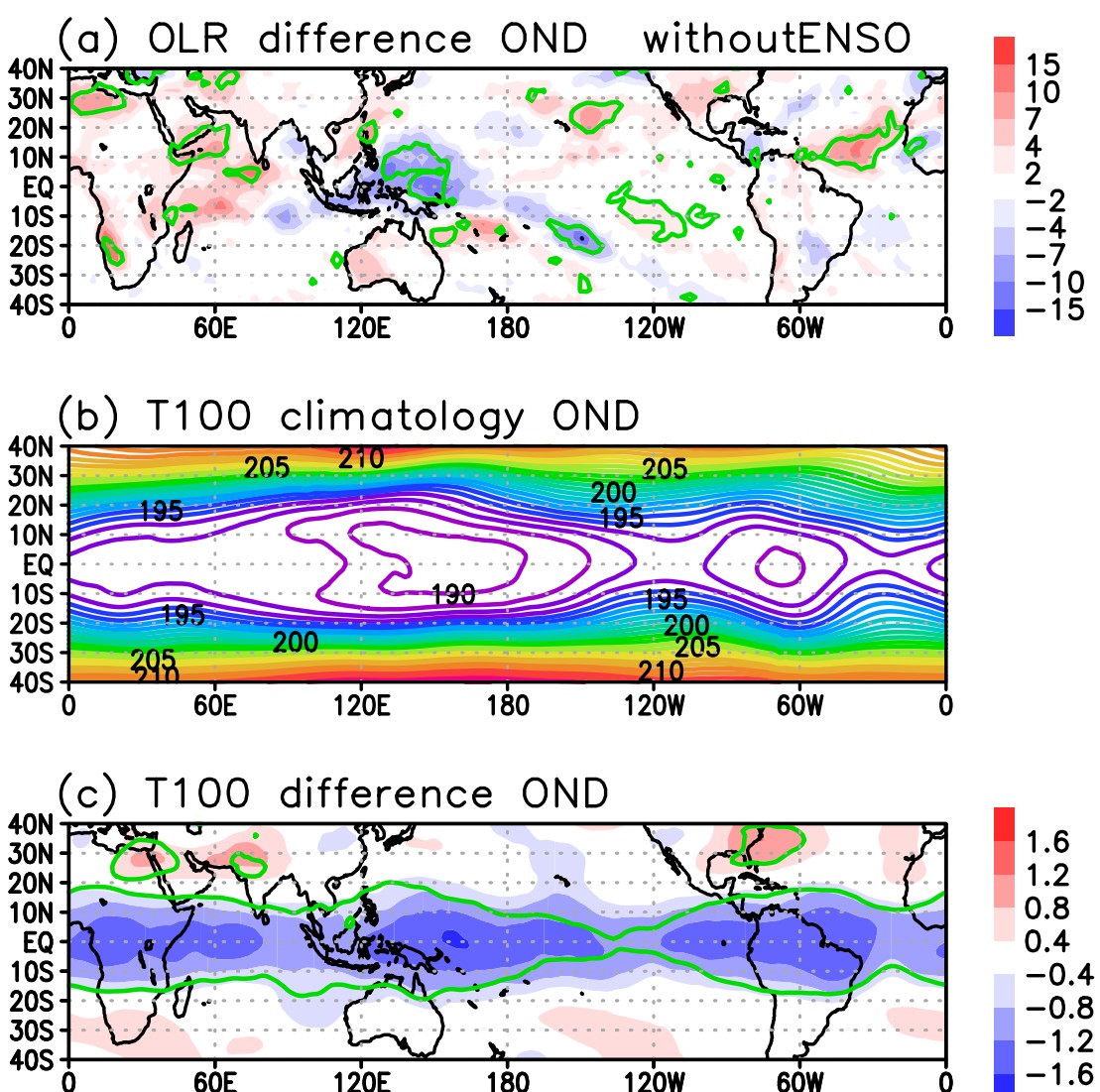

**Figure 4.** Same as Fig. 3 but without ENSO winters.

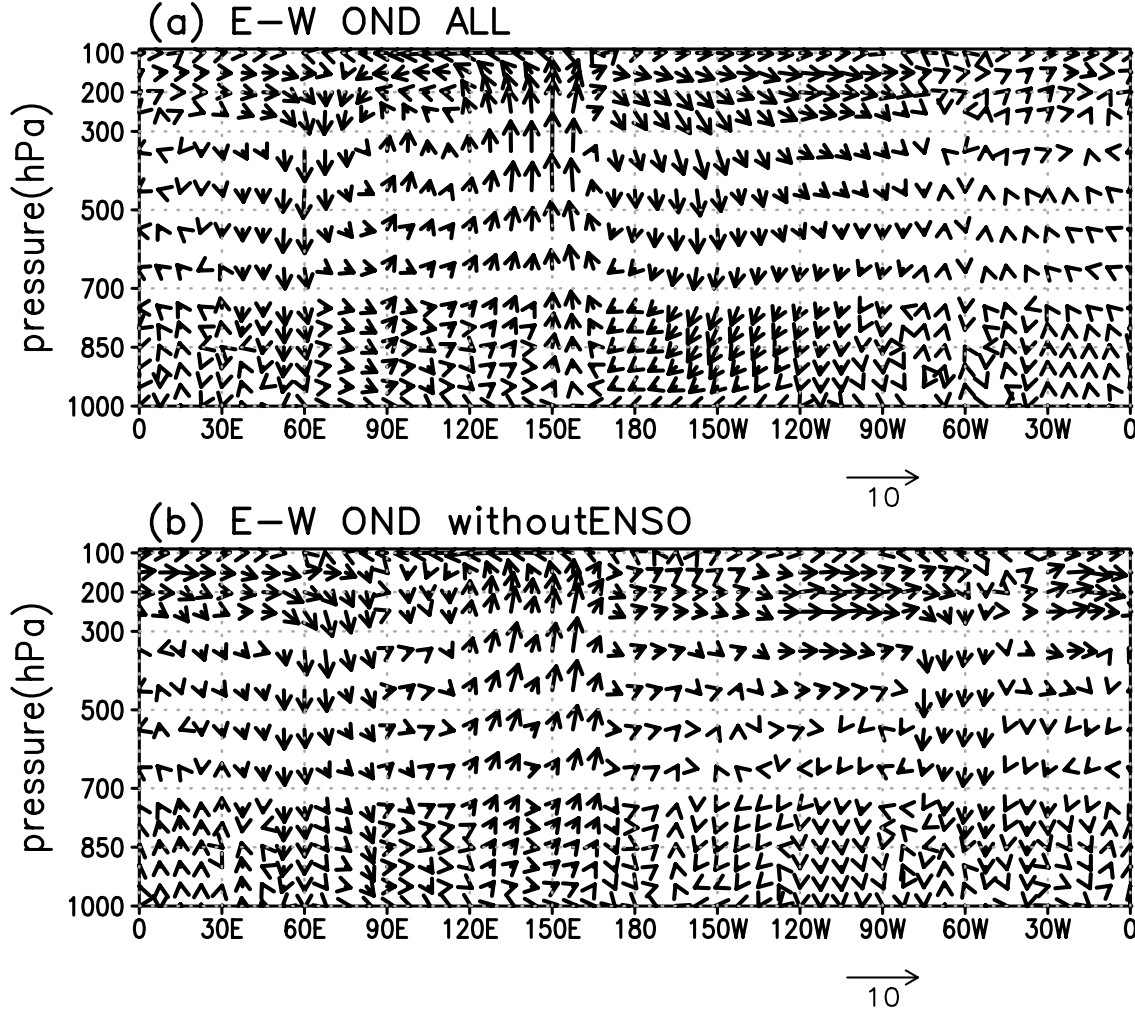

**Figure 5.** (a) October-November-December (OND) mean zonal-pressure circulation (u-wind and minus p-velocity) differences averaged from 10°S to 10°N between EQBO and WQBO winters. (b) Same as in (a) but without ENSO winters. Arrows below the figure show the scales for 10 m/s U-wind. p-velocity is multiplied by 300.

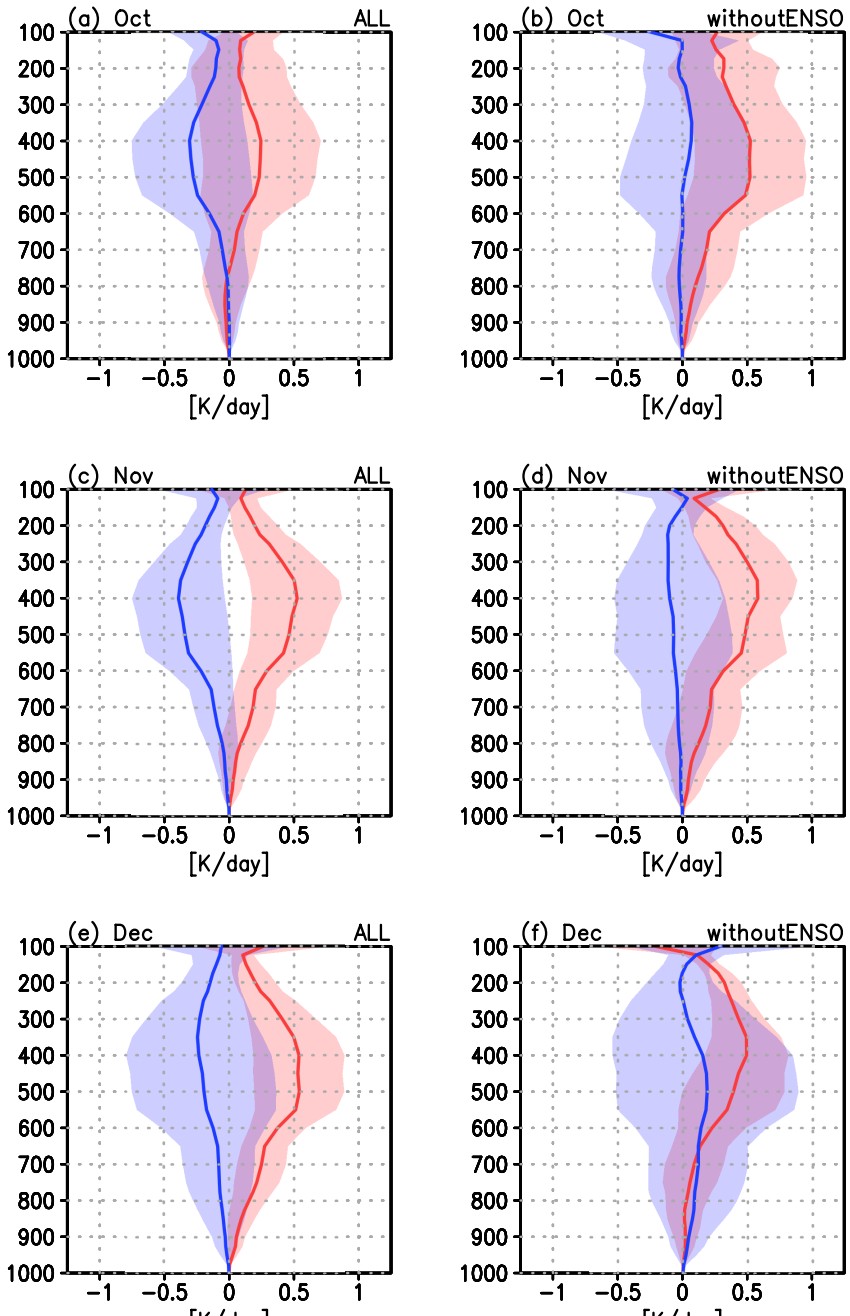

**Figure 6.** Vertical profiles of diabatic heating in temperature over western tropical Pacific region [130°-160°E, 0-10°N] derived from the conservation property of potential temperature (see Appendix B). Values are deviation from the climatology. Red lines show EQBO composite and blue lines show WQBO composite. Shadings show the twice of standard error, corresponding 95% confidence interval. (a) October with ENSO years. (b) October without ENSO years. (c) November with ENSO years. (d) November without ENSO years. (e) December with ENSO years. (f) December without ENSO years.

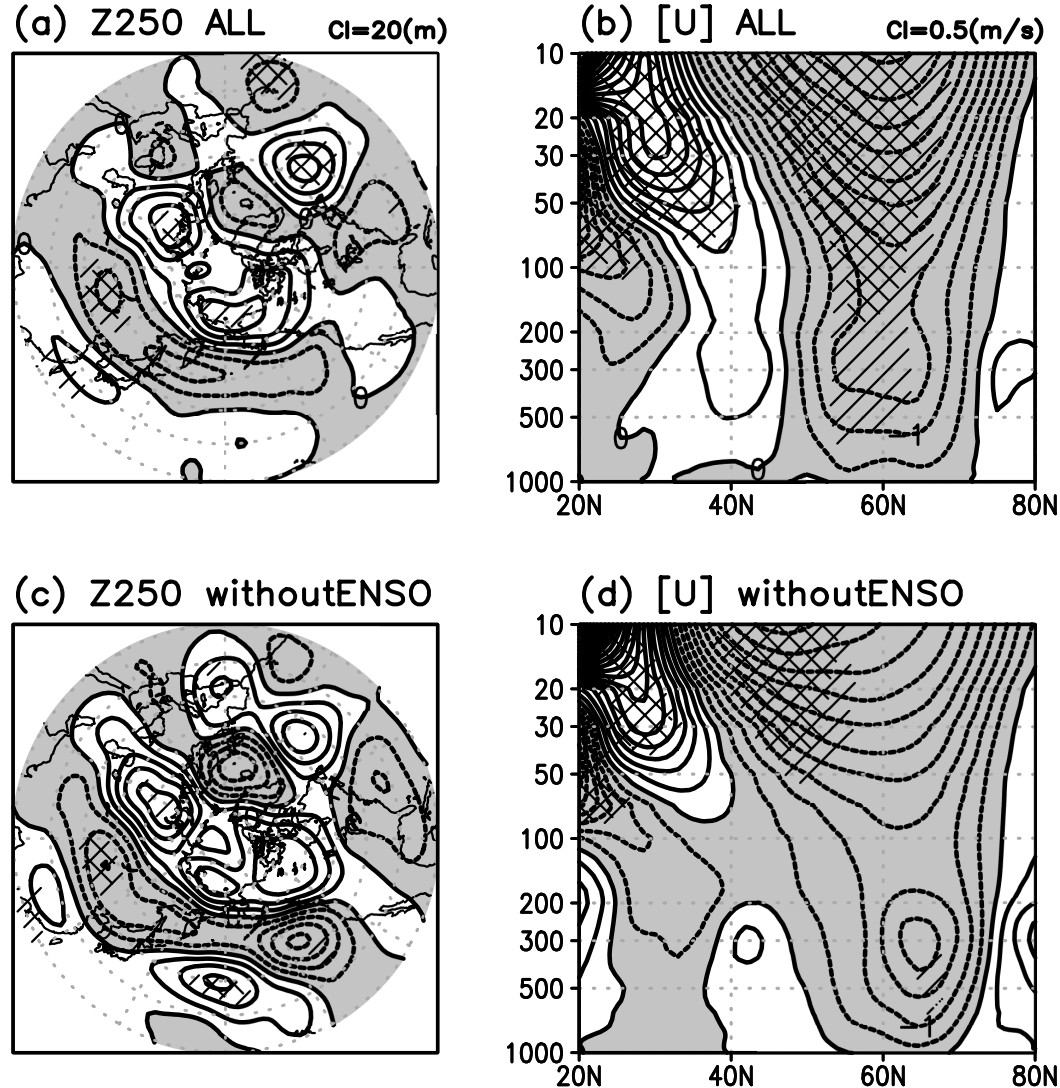

**Figure 7.** (a) Composite difference of geopotential height at 250 hPa (Z250) in November between EQBO and WQBO winters. Contour interval (CI) is 20 m and negative values are shaded. The values statistically significant at 90 and 95% level are hatched. (b) Same as in (a) but for the zonal mean zonal wind [CI= 0.5m/s]. (c) Same as (a) but without ENSO winters. (d) Same as (b) but without ENSO winters.

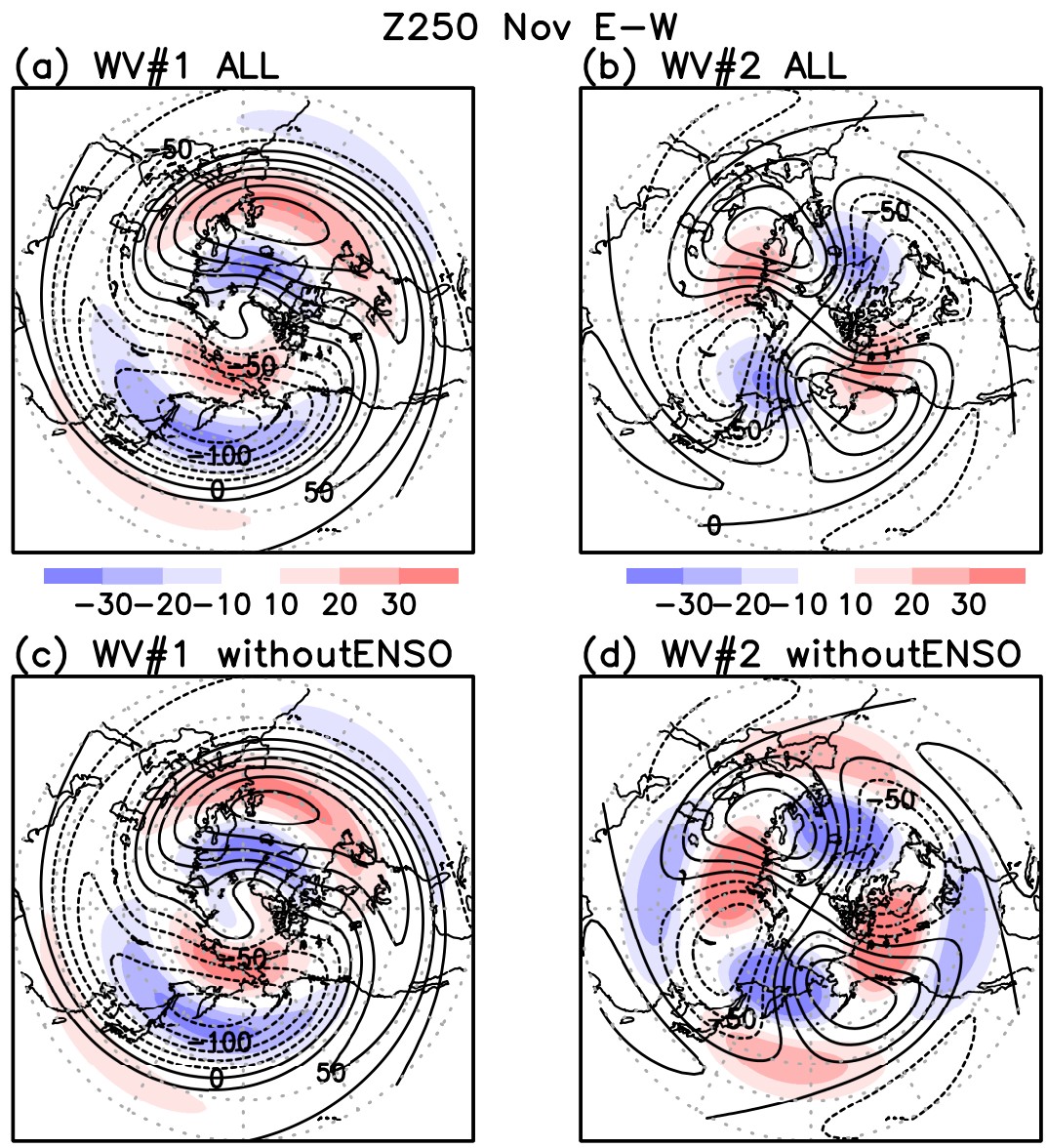

**Figure 8.** Composite difference of Z250 wavenumber 1 and 2 in November between EQBO and WQBO winters. (a) Z250 wavenumber 1 difference (shade) and climatological wavenumber 1 (contour). Contour interval is 25 m. (b) Same as (a) but for wavenumber 2. (c) Same as (a) but without ENSO winters. (d) Same as (c) but for wavenumber 2.



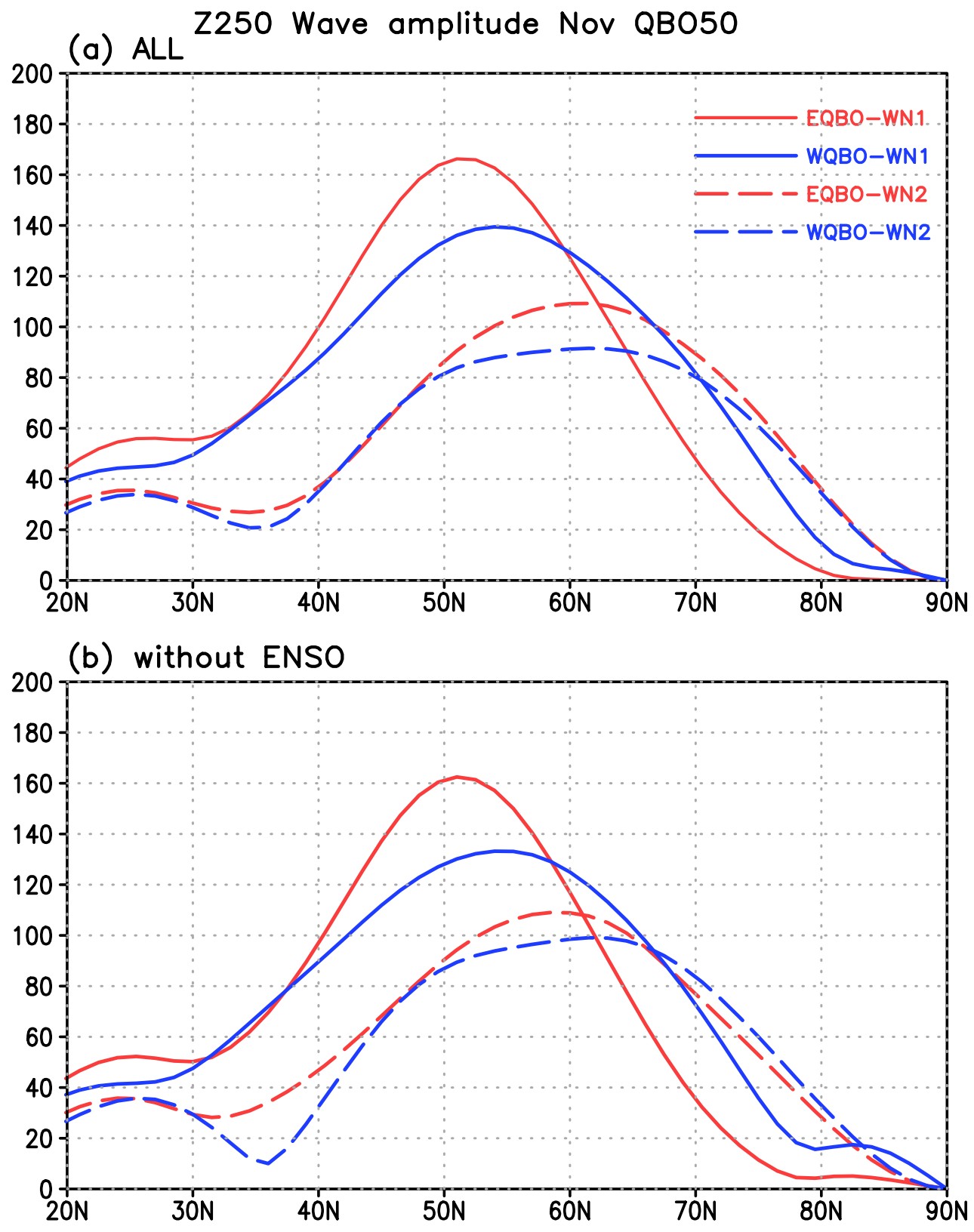

**Figure 9.** Wave amplitudes at 250 hPa as a function of latitude in the different QBO phases for November. Red (blue) solid line denotes wave-1 in the EQBO (WQBO) composite. Red (blue) dashed line denotes wave-2 in the EQBO (WQBO) composite. Y-axis denotes amplitude in m. (a) All composite. (b) Composite without ENSO winters.

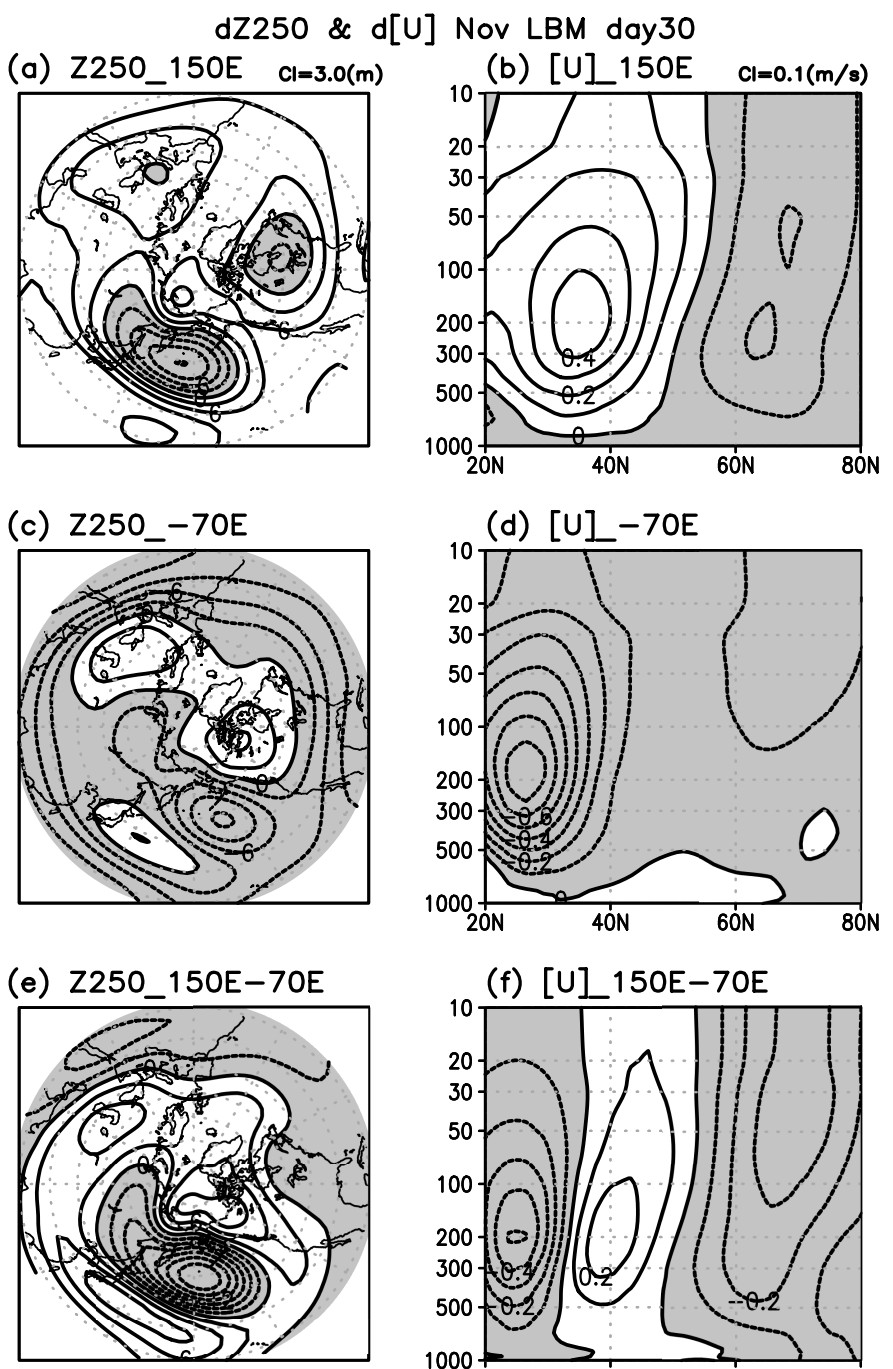

**Figure 10.** (a) Linear response of geopotential height at 250 hPa (Z250) to the adiabatic heating centered at 150°E, 5°N simulated by LBM with November climatological background field. Contour interval (CI) is 3 m. (b) Same as (a) but for response of the zonal mean zonal wind [CI= 0.1m/s]. (c) Same as in (a) but for the adiabatic cooling centered at 70°E, 5°N. (d) Same as (b) but for the adiabatic cooling centered at 70°E, 5°N. (e) (a)+(c). (f) (b)+(d). In all figures, negative values are shaded.



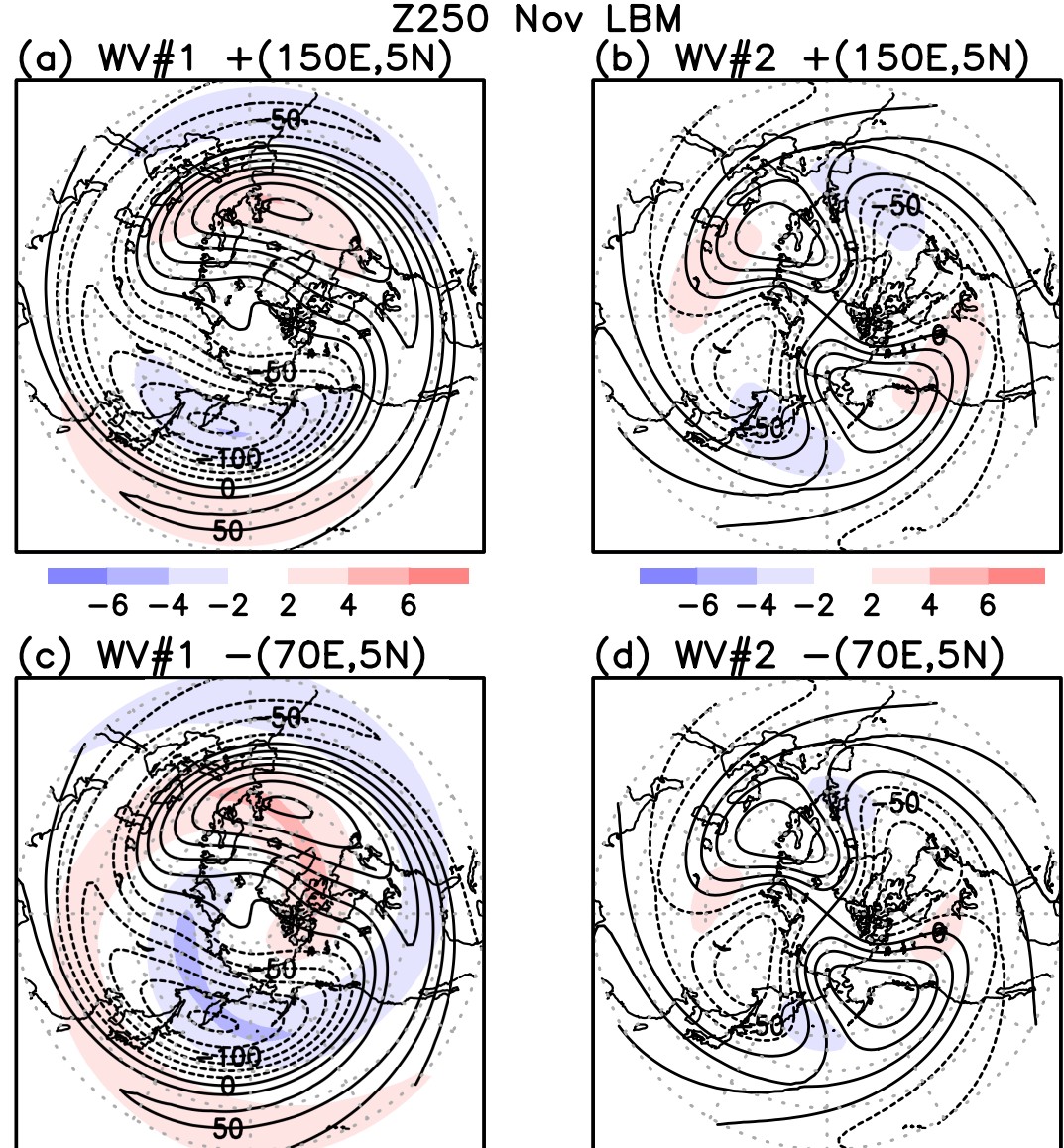

**Figure 11.** Wavenumber 1 and 2 components of linear responses of geopotential height at 250 hPa (Z250) simulated by LBM with November background field. (a) Z250 wavenumber 1 response at day 30 to the adiabatic heating centered at 150°E, 5°N (shade). Contour shows the corresponding November climatological wavenumber 1 field. Contour interval is 25 m. (b) Same as (a) but for wavenumber 2. (c) Same as (a) but for the adiabatic cooling centered at 70°E, 5°N. (d) Same as (c) but for wavenumber 2.



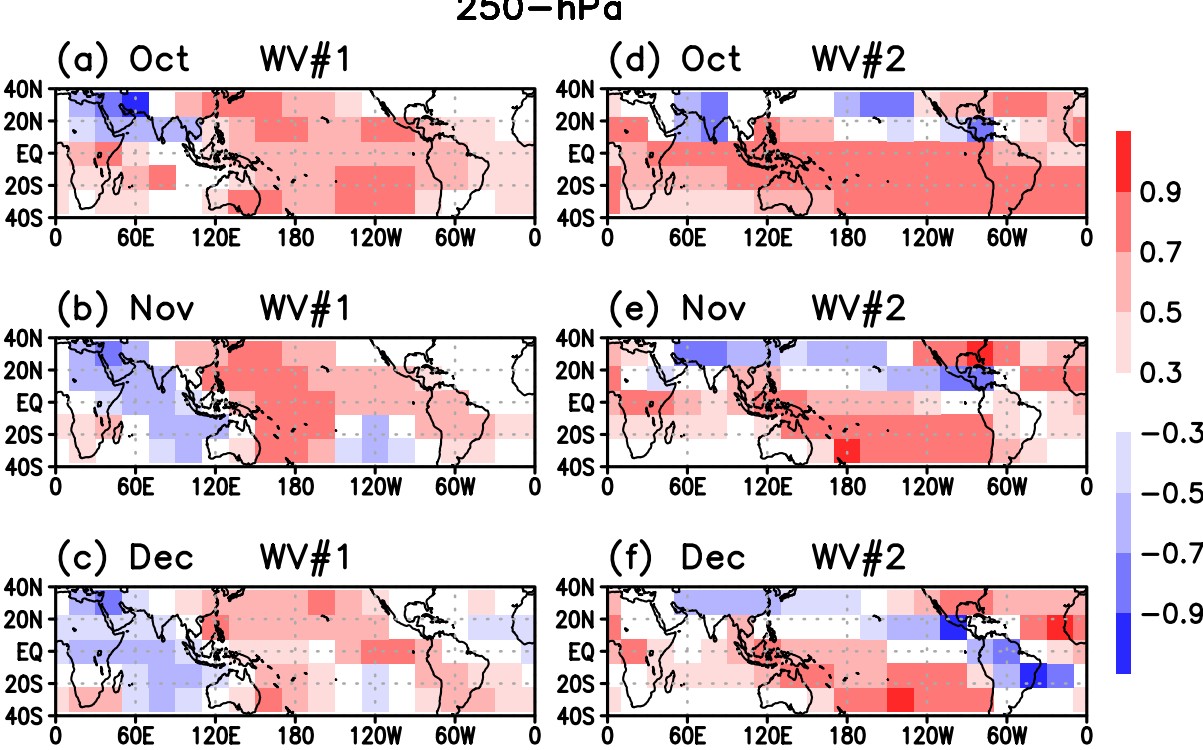


**Figure 12.** Correlation coefficients of spatial patterns between LBM-simulated eddy response to heating whose center is located at each grid and the climatological eddy height field north of 40°N at 250 hPa. Correlation value is plotted at the center of the heating location. (a)-(c) Wavenumber 1 field from October to December. (d)-(f) Same as (a)-(c), but for wavenumber 2 field.


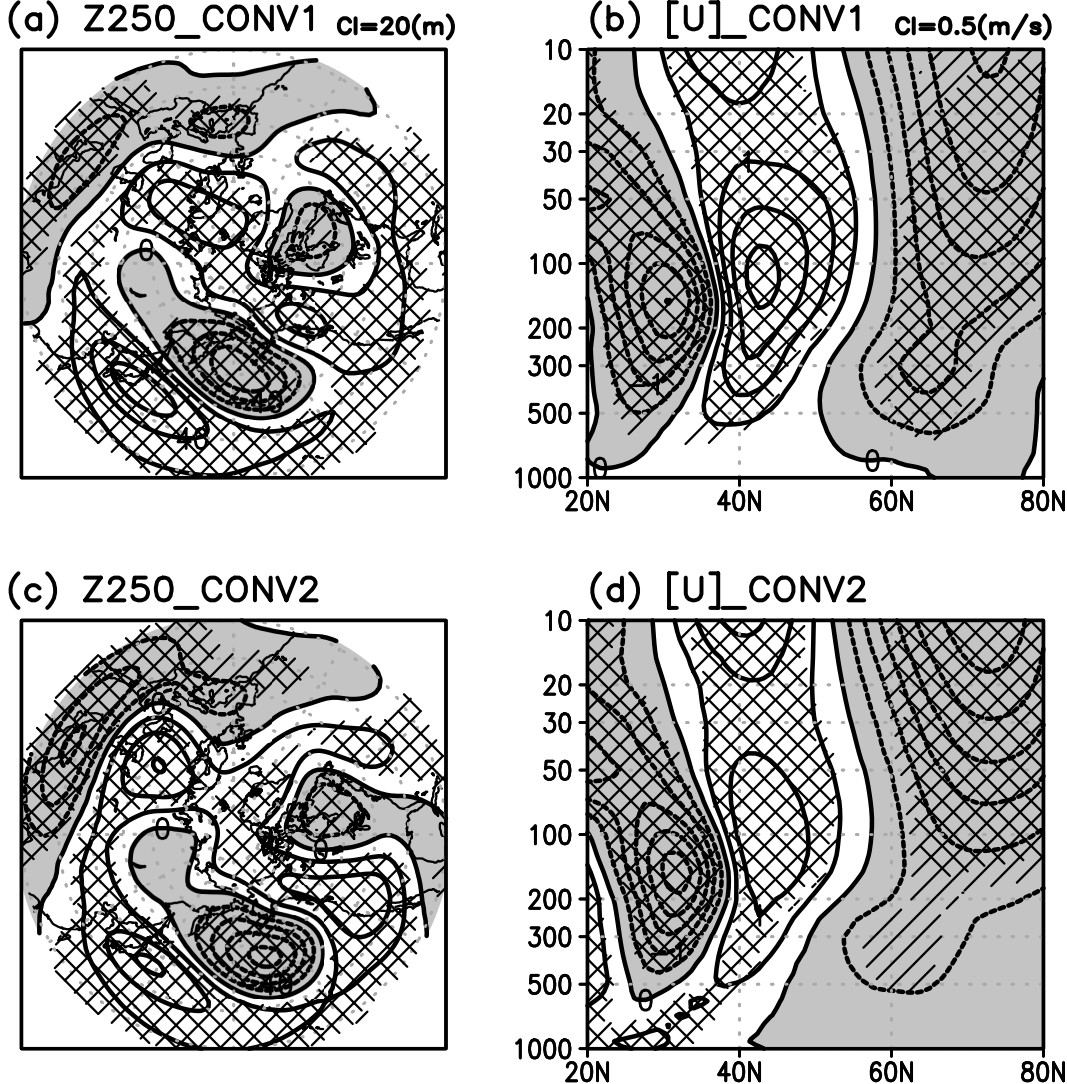

**Figure 13.** (a) AGCM simulated response of geopotential height at 250 hPa (Z250) in November to the diabatic heating centered at 150°E, 5°N against the control experiment. Contour interval (CI) is 20 m. The values statistically significant at 90 and 95% level are hatched. (b) Same as (a) but for response of the zonal mean zonal wind [CI= 0.5 m/s]. (c) Same as in (a) but for a pair of diabatic heating centered at 150°E, 5°N and cooling centered at 70°E, 5°N. (d) Same as (b) but for the pair of the heating and cooling. In all figures, negative values are shaded.


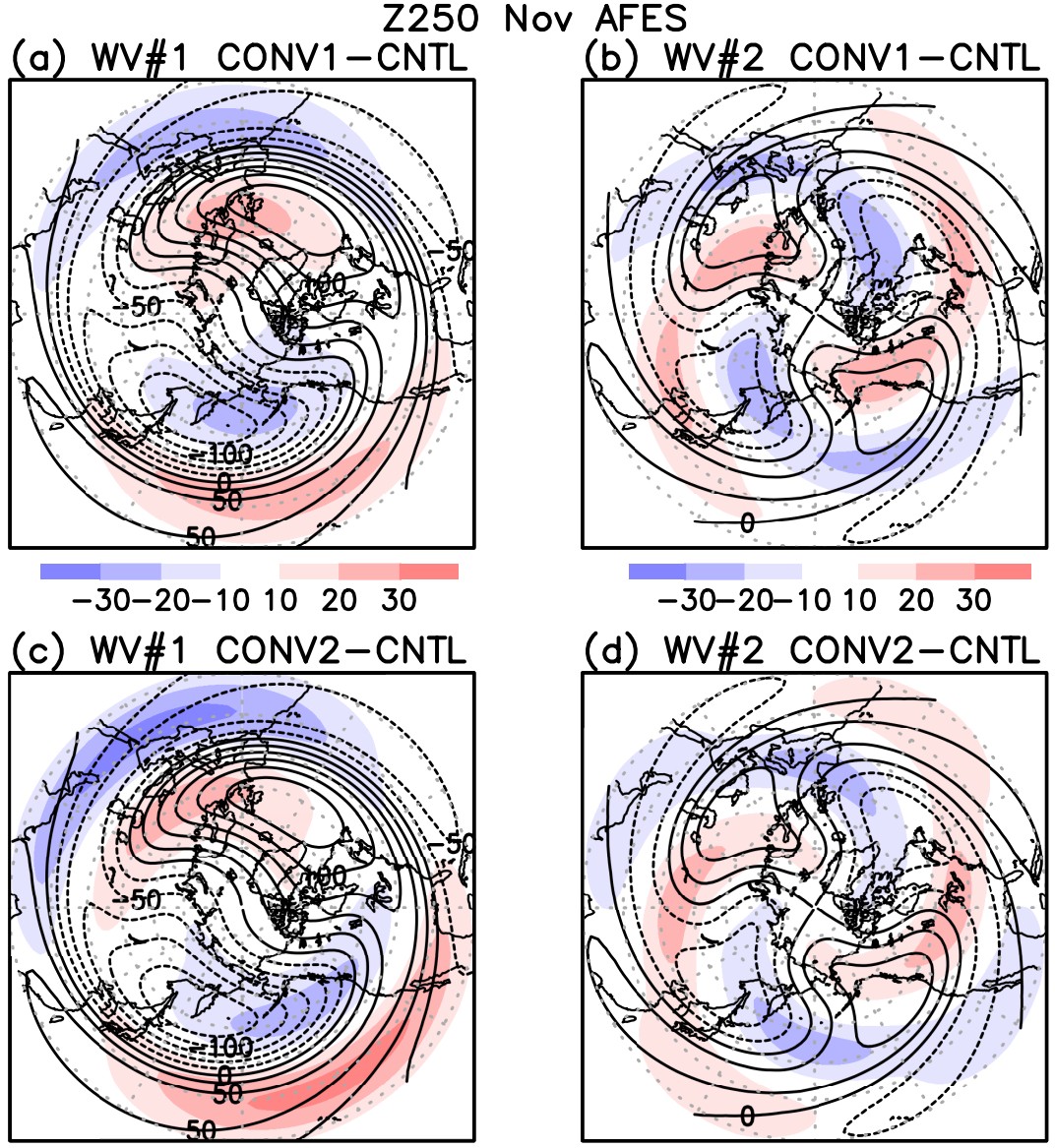

**Figure 14.** Wavenumber 1 and 2 components of AGCM simulated response of geopotential height at 250 hPa (Z250) in November. (a) Z250 wavenumber 1 response to the adiabatic heating centered at 150°E, 5°N (shade). Contour shows the corresponding November climatological wavenumber 1 field. Contour interval is 25 m. (b) Same as (a) but for wavenumber 2. (c) Same as (a) but for a pair of adiabatic heating centered at 150°E, 5°N and cooling centered at 70°E, 5°N. (d) Same as (c) but for wavenumber 2.

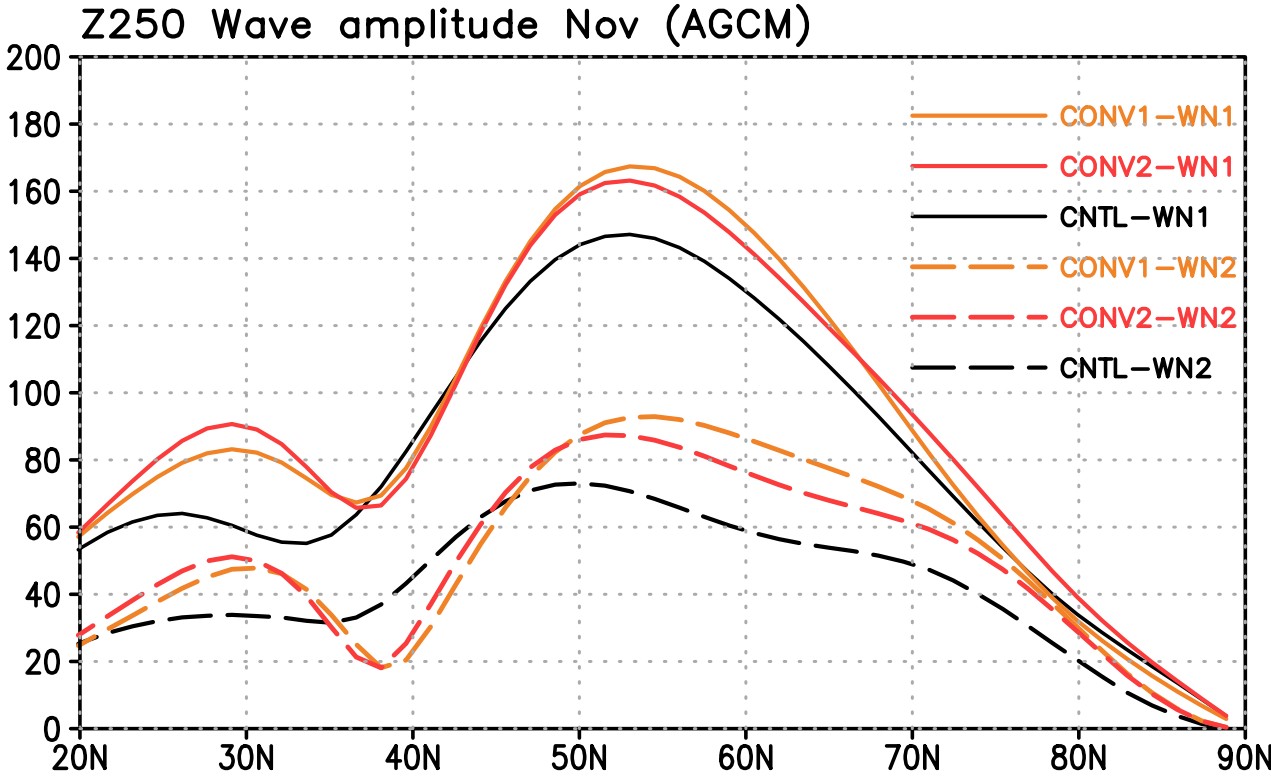

**Figure 15.** Latitudinal profile of the wave amplitudes at 250 hPa in AGCM simulations for November, based on 60-year mean. Black lines show the control simulation, orange lines for CONV1 simulation, and red lines for CONV2 simulation. Solid and dashed lines denote wave-1 and wave-2 components, respectively. Y-axis denotes amplitude in m.

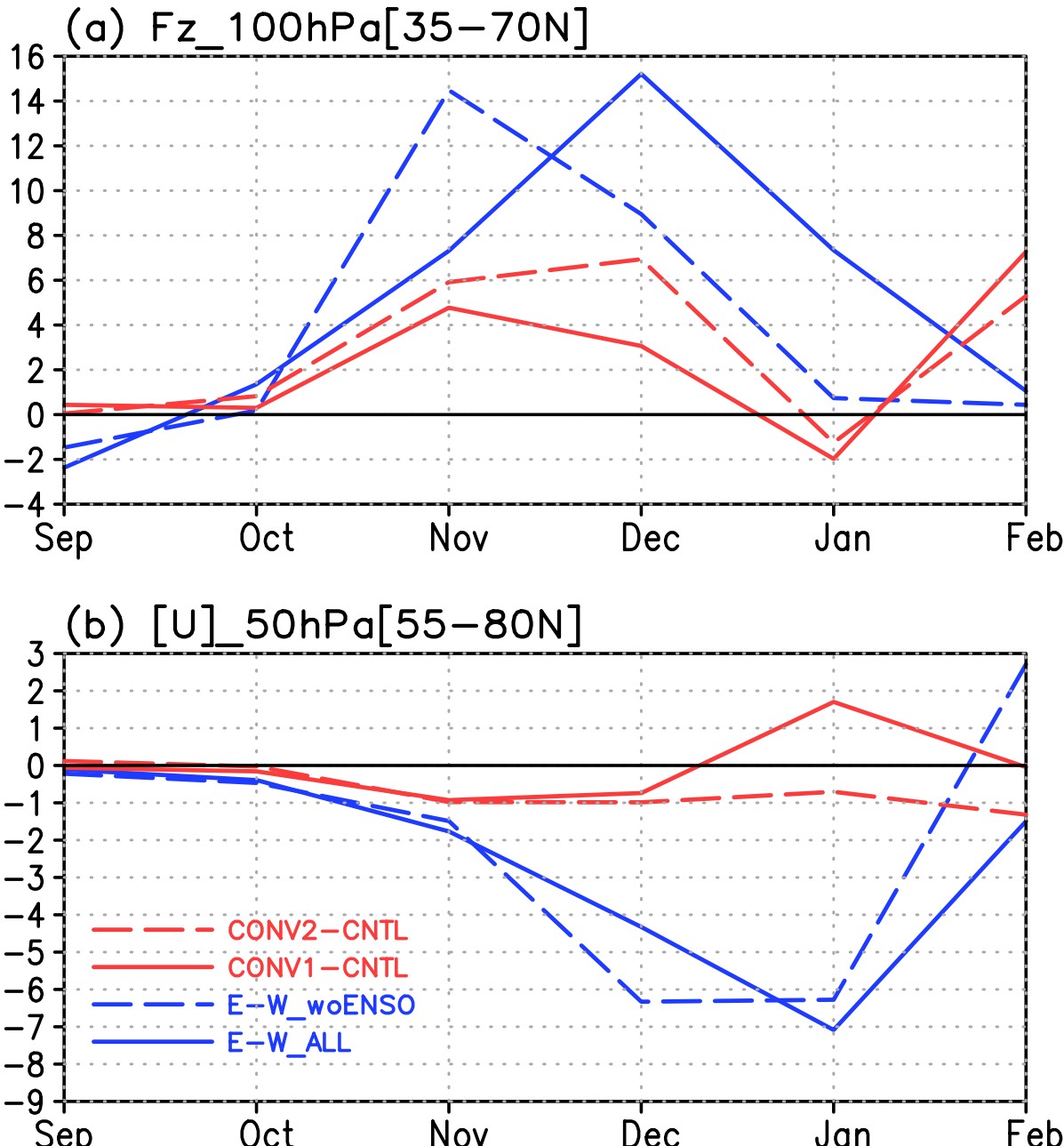


**Figure 16.** (a) Time series of upward EP flux (Fz) at 100 hPa averaged over 35°-70°N. Unit is $1\times10^3$ kg s$^{-2}$. Blue solid (dotted) line shows observed difference between EQBO and WQBO winters (without ENSO winters). Red solid (dotted) line shows difference between CONV1 (CONV2) and CNTL experiments. (b) Time series of zonal mean zonal wind [U] difference at 50 hPa averaged over 55°-80°N between EQBO and WQBO winters. Unit is m/s. Blue solid (dotted) line

shows the difference with all years (without ENSO winters). Red solid (dotted) line shows difference between CONV1 (CONV2) and CNTL experiments.


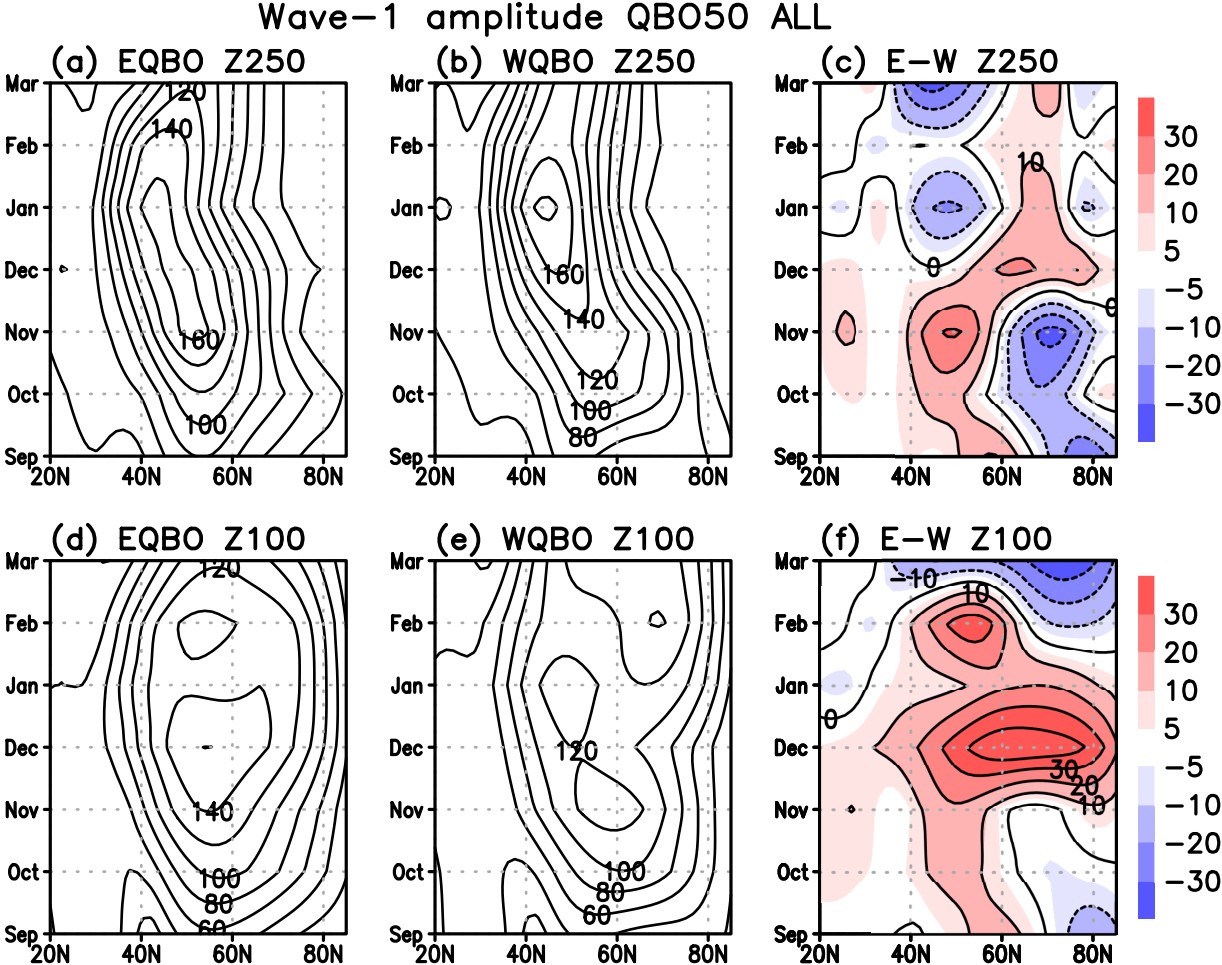

**Figure 17.** Latitude-month plots of observed wavenumber-1 amplitude. (a) EQBO composite at 250 hPa. (b) WQBO composite at 250 hPa. (c) Difference between EQBO and WQBO. (d)-(f) Same as in (a)-(c) but for 100 hPa. Contours in (a),(b),(d),and(e) are 20 m, and those in (c) and (f) are 10 m.

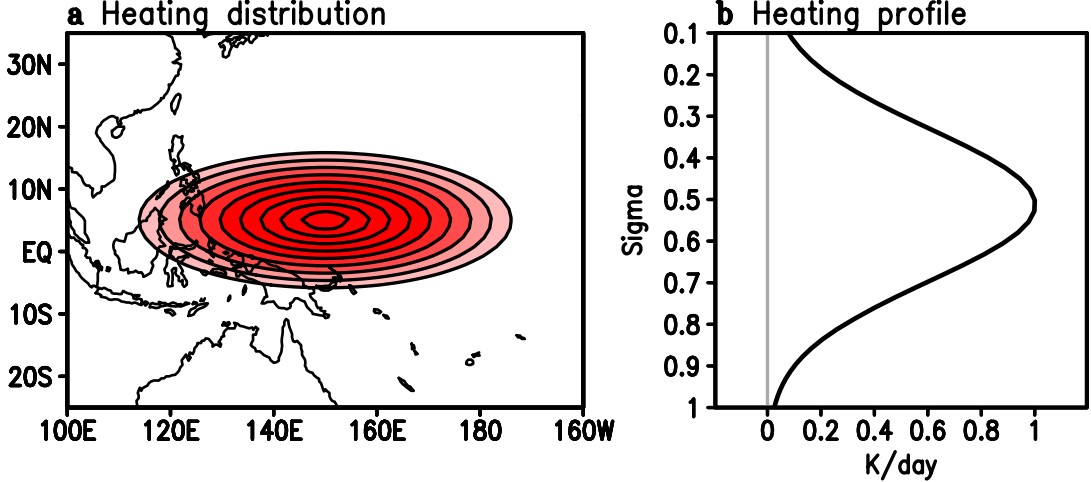

**Figure A1.** (a) Horizontal distribution of the heating given to force the LBM/AGCM at 0.5 sigma level. Contour is drawn from 0.1 K d$^{-1}$ to 0.9 K d$^{-1}$ with an interval of 0.1. (b) Vertical profiles of the heating at the location of maximum heating (150°E and 5°N).

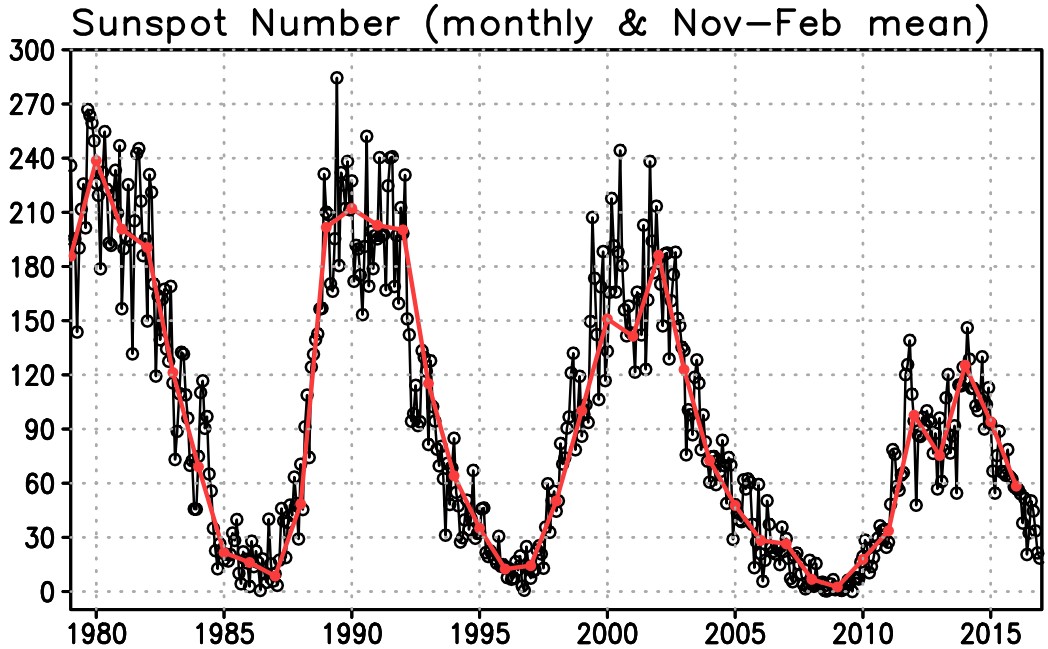

**Figure A2.** Monthly mean sunspot number (black with open circle) and November-February mean (red with closed circle).

**Table A.1.** Number of EQBO, WQBO, solar max, solar min and other years

|  | EQBO | WQBO | Others |
|---|---|---|---|
| Solar Max | 7 | 7 | 4 |
| Solar Min | 5 | 12 | 2 |

685

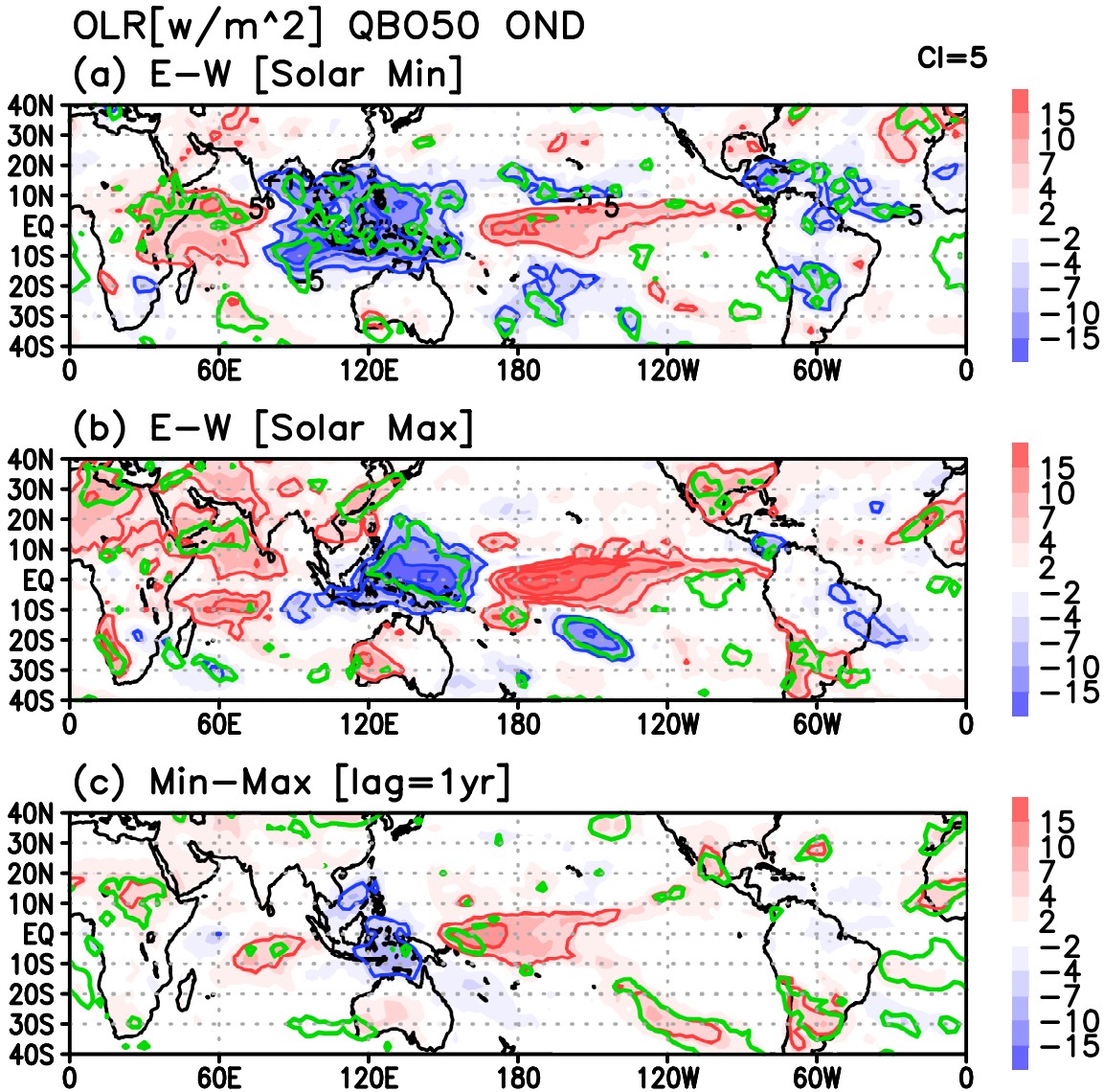

**Figure A3.** (a) October-November-December (OND) mean OLR differences between EQBO and WQBO winters for solar
690 minimum winters. (b) Same as (a) but for solar maximum years. (c) OND OLR differences between solar minimum and
maximum winters with 1-year lag. Green line denotes the statistically significant value at 95% confidence level.