# Peer review of "A tropospheric pathway of the stratospheric quasi-biennial oscillation (QBO) impact on the boreal winter polar vortex"

_Atmospheric Chemistry and Physics, 2019_

## Short Comment (SC1) · 17 Jan 2020

It would be useful to remind the reader that the QBO has a wavenumber=0, and therefore is distinct from off-equatorial effects. As with the upper stratospheric Semi-Annual Oscillation (SAO) which is clearly tidally forced by the solar semi-annual cycle, the QBO is forced by the group-symmetrical draconic/nodal lunar tide of 27.2122 days interacting with the annual signal. Only nodal tidal patterns are allowed for a wavenumber=0 standing wave.

---

## Author Comment (AC1) · 21 Jan 2020

Dear Dr. Paul Pukite, Thank you for your comment and interest in our paper. We agree that the QBO has a wavenumber=0. But, the QBO interacts with polar night jet through modulating planetary wave propagation as previous studies show (i.e., Baldwin et al., 2001). In the present study, we show that the QBO interacts with the tropical tropo-sphere that is zonally non-uniform. Hence the QBO produces non-zonally uniform convection anomalies. These convection anomalies cause off-equatorial effects through Rossby wave propagation. The QBO itself is forced by upward propagating equatorial waves such as Kelvin wave, mixed Rossby-gravity wave, and above all gravity waves

(Holton and Harkim, 2012 "Introduction to Dynamic Meteorology").

---

## Referee Comment (RC1) · Anonymous Referee #1 · 27 Jan 2020

**Review of Yamazaki et al. On Tropospheric Pathway for QBO Influence on Stratospheric Polar Vortex**

This paper presents evidence for the QBO having an influence on the stratospheric polar vortex (the Holton-Tan effect) via the troposphere, rather than via the stratosphere as in previous proposed mechanisms (though they show that other mechanisms are also at work, particularly in late winter). The idea is that the QBO-E stimulates convection in the tropical West Pacific and suppresses it in the western Indian Ocean, and these produce Rossby wave forcing that constructively interferes with the extratropical Rossby wave structure, and hence increases planetary wave forcing of the stratospheric vortex. This is based on observations of differences in tropical convective activity in different QBO phases and model simulations used to estimate the impact of this on extratropical wave activity. Overall I think the sequence of experiments holds together well and I like the tests done to check that the ENSO phase is not a strong confounding factor. I think a few more diagnostics are needed to show that the findings are robust.

Most significant comments:
1. Some diagnostics need to be shown to explain why the mechanism does not seem to occur from January onwards – has the apparent difference in tropical convection disappeared (which would suggest to me that it's not robust) or is the forcing of extratropical planetary waves not effective for some reason?
2. L149 – it would be helpful to give some explanation of why tropopause temperature anomalies associated with the QBO would "provide favorable conditions for enhanced convective activity", with references. Having confidence that the QBO really has the impact on tropical convection that is shown is crucial for believing the mechanism, so this would be useful.
3. The authors note that the impact of the QBO on tropical convection is associated with a changed strength of the Walker circulation. Misios et al. (2019, "Slowdown of the Walker circulation at solar cycle maximum", PNAS) found that there is an impact of the solar cycle on the Walker circulation, so this could be a confounding factor. It would make sense to check that the impact of the QBO on tropical convection found here is not dependent on the solar cycle phase.
4. What method is being used for the statistical significance tests? What assumptions are being made for this? What has been done to check the assumptions are reasonable? This should be clearly explained in the text. (Personally, I think using a bootstrap method is best as it requires relatively few assumptions to be made, but another method can be used if the assumptions behind it can be justified.)

Other comments:
1. L32-35 It should be made clear that the "Holtan-Tan mechanism" is just one proposed mechanism.
2. L80 The definition of ENSO phases used by the JMA should be given for clarity.
3. L90 It should be made clear that the "QBO signal" refers to the EQBO minus WQBO difference (rather than the difference from climatology).
4. L104 References are given claiming to show that "the performance of the model in the stratosphere is satisfactory", but I couldn't see an evaluation of the stratospheric performance in any of these references. Please point out where this can be found, give a reference showing this, show data yourselves or remove this remark. (I am not too bothered about the model's stratospheric performance on the whole – for simulating the wave forcing from the tropical convection anomaly, it is the tropospheric performance that matters most I

think, and it looks adequate from the figures given – I say this just so a lot of work is not done to validate the model's stratospheric performance.)

5. L134-5 What's the relevance of the remark about vertical motion near the Equator being downward?
6. L167-70 Plots of the wave amplitudes as a function of latitude in the different QBO phases would be helpful.
7. L183 It seems worth noting that the wave-1 response to the convective forcing shows a large signal in the North Atlantic that is not shown in the full response in fig.9. Is the right interpretation that higher wavenumbers are cancelling out the response in the North Atlantic?
8. L218-9 This could do with being more quantitative e.g. compare the simulated and observed wave-1 amplitude change averaged over 30-60N.
9. L223-6 The text could do with being clearer about which results are from observations/reanalysis. (This goes for other parts of the results section as well.)
10. Appendix B – is there a reference supporting this method?

---

## Referee Comment (RC2) · Anonymous Referee #2 · 29 Jan 2020

This paper presents the equatorial QBO influences on the Northern Hemisphere winter circulation. Many previous studies focused on the stratospheric pathways of this influences, while this manuscript proposes a possible mechanism for tropospheric pathways of this influences through the modulation of Rossby wave activities induced by the QBO-related convection over the tropical western Pacific and the Indian Ocean. This topic is interesting and valuable for this scientific area. However, there are some issues as mentioned below. For these reasons, I recommend minor revisions.

Minor comments:

(1) p.2, l.75: The definition of QBO is based on the absolute values of equatorial zonal

wind >3m/s. Please check another threshold and mention it.

(2) p.4, l.150: The difference between Fig. 5a and 5b indicates the influence of ENSO on the equatorial east Pacific as the downward around 150W with positive OLR in Fig. 3a. Is this interference from ENSO really ruled out in later analysis?

(3) p.5, l.l.165-170: Some references are needed for the constructive interference between the anomalous Rossby wave response and the background climatological stationary wave. Smith et al. (2010) showed the linear interference between these waves in their model. Using reanalysis data, Garfinkel et al. (2010) showed the constructive interference between the ENSO-related anomaly and climatology, and Yamashita et al. (2015) showed this interference between the QBO/solar-related anomaly and the climatology.

The constructive interference in Smith et al. (2010) is linear process, thus, it is reasonable that the constructive interference is reproduced with the LBM.

Garfinkel, C. I., D. L. Hartmann, and F. Sassi (2010), Tropospheric precursors of anomalous Northern Hemisphere stratospheric polar vortices, J. Climate, 23, 3282-3299.

Smith, K. L., C. G. Fletcher, and P. J. Kushner (2010), The role of linear interference in the annular mode response to extratropical surface forcing, J. Climate, 23, 6036-6050.

Yamashita, Y., H. Akiyoshi, T. G. Shepherd, and M. Takahashi (2015), The combined influences of westerly phase of the quasibiennial oscillation and 11-year solar maximum conditions on the Northern Hemisphere extratropical winter circulation, J. Meteor. Soc. Japan, 93, 629-644

(4) Some modifications of introduction are needed as mentioned below.

p.1, l.35: Holton and Tan, 1980, 1982 only show a plausible mechanism, as the latitudinal position of the zero-wind critical surface of stationary Rossby wave is primally controlled by the equatorial QBO. Recently, Watson and Gray (2014) posted this line

[Figure]

of discussion with their model.

Watson, P.A.G., and L.J. Gray (2014) How does the quasi-biennial oscillation affect the stratospheric polar vortex?, J. Atmos. Sci., 71, 391-409

p.1, l.35: "this critical latitude mechanism is not effective": The wave propagation change between the EQBO and WQBO is similar to the previous studies in high-latitudes and around equator in Naoe and Shibata (2010)'s results, in agreement with Holton-Tan relationship. In contrast, another propagation change, which is opposite to the critical line control, is analyzed in mid-latitudes by Naoe and Shibata (2010). White et al. (2015) suggested the enhanced upward wave propagation at midlatitudes due to the enhanced wave growth rather critical latitude mechanism, explaining the QBO-related change in mid-latitudes as well as the polar vortex change in high-latitudes.

White, I.P., H. Lu, N.J. Mitchell, and T. Phillips (2015), Dynamical response to the QBO in the northern winter stratosphere: Signatures in wave forcing and eddy fluxes of potential orticity. J.Atmos. Sci., 72, 4487-4507.

p.1, l.35: "The secondary circulation associated with the QBO in the subtropics": Naoe and Shibata (2010) and Yamashita et al. (2011) suggested the significance of the secondary circulation induced by the equatorial QBO in middle stratosphere rather lower stratosphere. In contrast, Garfinkel et al. (2012) and Lu et al. (2014) suggested the significance of the QBO-induced meridional circulation anomalies extend from the subtropics to the midlatitudes in relation to the midlatitudes change of Rossby waves due to the changes in index of refraction.

Other comments:

p.4, l.155: The middle tropospheric values of red lines (WQBO) are positive and the blue lines (EQBO) are negative in Fig. 6. Does it indicate the relatively large diabatic heating in the WQBO?

p.6, l.210: Fig. 9a shows the dipole pattern between mid-latitudes and Polar region,

while Fig. 12a shows the tri-pole pattern.

p.5, l.200: I suppose that "no interaction between the anomalous response and climatological fields" in terms of nonlinear processes, since the LBM model has the constructive interference for linear processes only.

p.6, l.215: I suppose that the constructive interference is valid, when the anomalous waves and climatological waves are in phase, as the description of wavenumber 1 field at p.5, l.170. But, their wavenumber 2 fields in Fig.8 are out of phase.

Typos: p.1, l.35: atmospheric general circulation models (AGCMs)

p.4, l.145: Fig. 4c, 5c -> Fig. 3c, 4c

---

## Short Comment (SC2) · 29 Jan 2020

The solar influence is likely not the 11 year sunspot cycle, but the annual and semi-annual cycle. That is obvious because the semi-annual oscillation (SAO) at altitudes above the QBO are obviously semi-annual.

---

## Author Comment (AC2) · 6 Mar 2020

**Review of Yamazaki et al. On Tropospheric Pathway for QBO Influence on Stratospheric Polar Vortex**

This paper presents evidence for the QBO having an influence on the stratospheric polar vortex (the Holton-Tan effect) via the troposphere, rather than via the stratosphere as in previous proposed mechanisms (though they show that other mechanisms are also at work, particularly in late winter). The idea is that the QBO-E stimulates convection in the tropical West Pacific and suppresses it in the western Indian Ocean, and these produce Rossby wave forcing that constructively interferes with the extratropical Rossby wave structure, and hence increases planetary wave forcing of the stratospheric vortex. This is based on observations of differences in tropical convective activity in different QBO phases and model simulations used to estimate the impact of this on extratropical wave activity. Overall I think the sequence of experiments holds together well and I like the tests done to check that the ENSO phase is not a strong confounding factor. I think a few more diagnostics are needed to show that the findings are robust.

We appreciate Reviewer #1 very much for the constructive comments and suggestions. We have carefully incorporated comments and suggestions, which have improved the manuscript in its content and presentation. Our responses to the specific comments can be found below in black (Reviewer #1's comments and suggestions) and blue (our responses).

Most significant comments:

1. Some diagnostics need to be shown to explain why the mechanism does not seem to occur from January onwards – has the apparent difference in tropical convection disappeared (which would suggest to me that it's not robust) or is the forcing of extratropical planetary waves not effective for some reason?

> Thank you. We think the stratospheric pathway is effective in midwinter, in short. We made an analysis on wave amplitudes and the results are now included in the revised version. The following sentences are included in the revised version.

"Why the tropospheric mechanism does not seem to occur from January onward? Is this because QBO-induced tropical convection anomalies disappear in midwinter? We examined the observed tropical convection difference between EQBO and WQBO for each month and found that it does not disappear but shifts slowly eastward. We then made simple diagnostics on seasonal change in observed wave amplitudes (Fig. 17). At 250 hPa, from September to November, wave-1 amplitude in EQBO is larger than that in WQBO at around the maximum latitude. This means that the maximum wave-1 amplitude is enhanced. In December, wave-1 amplitude in EQBO is enhanced in high-latitudes. Although this high-latitude enhancement continues to March, wave-1 amplitude at the maximum latitude of 50°N is reduced and no wave-1 amplitude enhancement in the troposphere is seen from January onward. On the contrary, the stratospheric polar vortex in EQBO weakens more in January (Figs. 1a and 16b). This corresponds to an enhancement of wave-1 amplitude at 100 hPa (Fig. 17f). Apparently, wave-1 amplitude in EQBO becomes larger than that in WQBO from November to February. For wave-2, the seasonal march at 100 hPa and that at 250 hPa are similar (not shown). We suppose the stratospheric processes discussed in many previous studies can account for the mid-winter Holton-Tan relationship. In mid-winter, the stratosphere undergoes vacillation

without changes in the troposphere (Holton and Mass, 1976; Chen et al., 2001; de la Cámara et al., 2019). "

[Figure]

**Figure 17.** Latitude-month plots of observed wavenumber-1 amplitude. (a) EQBO composite at 250 hPa. (b) WQBO composite at 250 hPa. (c) Difference between EQBO and WQBO. (d)-(f) Same as in (a)-(c) but for 100 hPa. Contours in (a),(b),(d),and(e) are 20 m, and those in (c) and (f) are 10 m.

Holton, J. R., and C. Mass, 1976: Stratospheric vacillation cycles, J. Atmos. Sci., 33, 2218-2225, https://doi.org/10.1175/1520-0469(1976)033<2218:SVC>2.0CO;2.

Chen, M., Mechoso, C. R., and Farrara, J. D.: Interannual variations in the stratospheric circulation with a perfectly steady troposphere, J. Geophys. Res., 106, 5161-5172, https:// di.org/10.1029/2000JD900624, 2001.

de la Cámara, A., T. Birner, and J. R. Albers, 2019: Are sudden stratospheric warmings preceded by anomalous tropospheric wave activity? J. Climate, 32, 7173-7189. DOI:10.1175/ JCLI-D-19-0269.1.

2. L149 – it would be helpful to give some explanation of why tropopause temperature anomalies associated with the QBO would "provide favorable conditions for enhanced convective activity", with references. Having confidence that the QBO really has the impact on tropical convection that is shown is crucial for believing the mechanism, so this would be useful.

> Thank you. Although we do not know precise mechanisms by which negative tropopause temperature anomalies provide favorable conditions for enhanced convective activity, we suspect weak stability and subsequent increase in cloudiness in the tropical tropopause layer (TTL) are the

main two key elements. In addition feedback arising from cooling in the TTL and warming in the mid-troposphere by cloud longwave forcing may farther accelerate weak stability, thereby enhancing the convective activity, as noted by Giorgetta et al. (1999) for boreal summer season.

So we added the Giorgetta et al (1999) in the references and explanation is added in the revised version.

Peña-Ortiz et al (2019) examined QBO influence the tropical convection and showed QBO modulation of the tropical convection that impacts stationary waves and the polar vortex of the austral winter of the southern hemisphere.
We added this paper in Introduction and references.

Giorgetta, M. A., Bengtsson, L., and Arpe, K.: An investigation of QBO signals in the East Asian and Indian monsoon in GCM experiments, Climate Dyn., 15, 435-450, 1999.
Peña-Ortiz, C., Manzini, E., and Giorgetta, M.: Tropical deep convection impact on southern winter stationary waves and its modulation by the Quasi-Biennial Oscillation, J. Climate, 32, 7453-7467. DOI: 10.1175/JCLI-D-18-0763.1, 2019.

3. The authors note that the impact of the QBO on tropical convection is associated with a changed strength of the Walker circulation. Misios et al. (2019, "Slowdown of the Walker circulation at solar cycle maximum", PNAS) found that there is an impact of the solar cycle on the Walker circulation, so this could be a confounding factor. It would make sense to check that the impact of the QBO on tropical convection found here is not dependent on the solar cycle phase.

> Thank you for the comment on solar cycle. Given possible compounding influences we further examined solar cycle modulation of the QBO impact on tropical convection found in our paper. The following is the results of our analysis which show only small impacts from the solar cycle and the robustness of our main conclusions. We added the analysis in the revised version as Section 3.7 and figures in Appendix as Figs. A2 and A3.

For the analysis we used time-series of the monthly and Nov-Feb mean sunspot number. The sunspot number data have been obtained from the World Center for Sunspot Index and Long-term Solar Observation (WDC-SILSO), Royal Observatory of Belgium, Brussels (http://www.sidc.be/silso/datafiles; Clette et al. 2014).

We added Section 3.7 as follows

**3.7 Modulation by 11-year solar cycle**

It has been known that the Holton-Tan relation is modified by the 11-year solar cycle (Labitzke, 2005, and references therein). Recently, Misios et al. (2019) provided strong evidence for weakened Walker circulation at the solar maximum. Recognizing possible compounding influences by the solar cycle on the QBO impact on tropical convection and extra-tropical circulation anomalies as discussed in our paper so far, we have made additional composite analysis as follows.

We used the Nov-Feb mean sunspot number as a solar index (SSN; Fig. A2), whose average value is 92.2. Winters above (SSN>92.2) and below (SSN<92.2) the average are classified as solar

max and solar min winters, respectively. We also divided winters into EQBO, WQBO composites, and other winters as described in Section 2.2 (see Table A1 for the sample size of each category). As identified in Misios et al. (2019), the solar impacts on convective activity thus the Walker circulation have one to two years of time lag through the bottom-up mechanism. We thus shifted by one year when classifying solar max and min winters. This sampling scheme provides consistent results with theirs on the solar influence, i.e. stronger Walker circulation at solar minimum seen in Figure A3c.

The QBO signal (EQBO minus WQBO) in OLR is stronger in the solar min years with significantly enhanced convection over the western tropical Pacific. In the solar max years, enhanced convection over the western tropical Pacific is weaker and shifts eastward slightly. Despite some differences, the QBO signal characterized by enhanced convection in the western tropical Pacific is commonly found in both solar max and min composites.

[Figure]

**Figure RC1.1. (new Fig. A2)** Monthly mean sunspot number (**black** with open circle) and November-February mean (red with closed circle).

**Table RC1.1.** Number of EQBO, WQBO, Solar max, solar min and other years

|            | EQBO | WQBO | Others |
|------------|------|------|--------|
| Solar Max  | 7    | 7    | 4      |
| Solar Min  | 5    | 12   | 2      |

[Figure]

**Figure RC1.2. (new Fig. A3)** (a) October-November-December (OND) mean OLR differences between EQBO and WQBO winters for solar minimum winters. (b) Same as (a) but for solar maximum years. (c) OND OLR differences between solar minimum and maximum winters with 1-year lag. Green line denotes the statistically significant value at 95% confidence level.

References
Clette, F., L. Svalgaard, J. M. Vaquero, and E. W. Cliver, 2014:Revisiting the sunspot number: A 400-year perspective on the solar cycle. Space Sci. Rev., 186, 35–103.

Labitzke, K., 2005: On the solar cycle-QBO relationship: a summary, J. Atmos. Solar-Terr. Phys. 67, 45-54.

Misios, S., L. J. Gray, M. F. Knudsen, C. Karoff, H. Schmidt, and J. D. Haigh, 2019: Slowdown of the Walker circulation at solar cycle maximum, PNAS, 116, 7186-7191.

4. What method is being used for the statistical significance tests? What assumptions are being made for this? What has been done to check the assumptions are reasonable? This should be

clearly explained in the text. (Personally, I think using a bootstrap method is best as it requires relatively few assumptions to be made, but another method can be used if the assumptions behind it can be justified.)

> Thank you. Throughout the manuscript we used t-statistics for all significant tests. Data is assumed to be normally distributed with no significant serial correlation. For example, auto-correlation of the OND averaged OLR over the western tropical Pacific (130-160E, 0-10N) with 1-year time lag is -0.23. Since the variation in this quantity is so central to our claim on the QBO impacts we examined its distribution (see Fig. RC1.3.) for EQBO and WQBO. The mean OND-averaged OLR for EQBO is 218.6 $W/m^2$, and 229.2 $W/m^2$ for WQBO where the average for whole 37 years is 224.6 $W/m^2$. We further employed the bootstrap method (n=1000000) to test its significant. The p-value is 99.97% against the observed difference of 10.58 $Wm^{-2}$.

[Figure]

**Figure RC1.3.** Histgram of western tropical Pacific OLR for October-December mean. The interval of bin is 4 $W/m^2$. Red line denotes EQBO winters, blue line denotes WQBO winters, and black dashed line denotes other winters.

Other comments:

1. L32-35 It should be made clear that the "Holtan-Tan mechanism" is just one proposed mechanism.

> Thank you. We modified the sentence following your suggestion and reviewr#2's suggestion.

"Holton and Tan (1980, 1982) only showed a plausible mechanism, as the latitudinal position of the zero-wind critical surface of stationary Rossby wave is primarily controlled by the equatorial QBO. Recently, Watson and Gray (2014) posted this line of discussion with their model. Naoe and Shibata (2010) analyzed Holton-Tan relationship by a QBO-producing chemistry-climate model (CCM) and

reanalysis data. They showed the conventional critical latitude mechanism that the equatorial winds in the lower stratosphere acted as a waveguide for planetary wave propagation did not hold. White et al. (2015) suggested the enhanced upward wave propagation at mid-latitudes due to the enhanced wave growth rather than the critical latitude mechanism, explaining the QBO-related change in mid-latitudes as well as the polar vortex change in high-latitudes."

2. L80 The definition of ENSO phases used by the JMA should be given for clarity.

>Thank you. "The definition of ENSO used by the JMA is based on 5-month moving averaged SST deviation from the standard value at NINO.3 (5°S-5°N, 150°W-90°W). When the SST deviation experiences more (less) than +0.5K(-0.5K) over 6 consecutive months, it is defined as El Niño (La Niña). The standard value is defined by previous 30-year mean for each month."

   Above sentences has been added. The following is the figure from JMA for reference.

[Figure]

   El Nino (pink) and La Nina (blue) periods defined by the JMA.

3. L90 It should be made clear that the "QBO signal" refers to the EQBO minus WQBO difference (rather than the difference from climatology).

> Thank you. We added the following sentence.

"Here, the QBO-signal refers to the EQBO minus WQBO difference rather than the deviation from climatology."

4. L104 References are given claiming to show that "the performance of the model in the stratosphere is satisfactory", but I couldn't see an evaluation of the stratospheric performance in any of these references. Please point out where this can be found, give a reference showing this, show data yourselves or remove this remark. (I am not too bothered about the model's stratospheric performance on the whole – for simulating the wave forcing from the tropical convection anomaly, it is the tropospheric performance that matters most I think, and it looks adequate from the figures given – I say this just so a lot of work is not done to validate the model's stratospheric performance.)

> Thank you. The stratospheric performance does not matter in this paper. So we deleted the sentence. Also we deleted references (Jaiser et al 2016 and Hoshi et al 2019).

5. L134-5 What's the relevance of the remark about vertical motion near the Equator being downward?

> We would like to have shown the QBO-signal is not zonally uniform. But, it is not so relevant and showing Figs.3a and 4a is enough. So we deleted the sentence. Thank you.

6. L167-70 Plots of the wave amplitudes as a function of latitude in the different QBO phases would be helpful.

> Thank you very much for the suggestion.

Latitudinal profiles of wave-1 and wave-2 amplitudes for EQBO and WQBO are shown below (Figure RC1.4; new Figure 9). "Peak values of the wave amplitude in November increase in EQBO for both wave-1 and wave-2 and regardless of in/exclusion of ENSO years." The figure and above sentence are added in the revised version. Also description for wave-2 was modified.

[Figure]

**Figure RC1.4. (new Fig. 9)** Wave amplitudes at 250 hPa as a function of latitude in the different QBO phases for November. Red (blue) solid line denotes wave-1 in the EQBO (WQBO) composite. Red (blue) dashed line denotes wave-2 in the EQBO (WQBO) composite. Y-axis denotes amplitude in m. (a) All composite. (b) Composite without ENSO winters.

7. L183 It seems worth noting that the wave-1 response to the convective forcing shows a large signal in the North Atlantic that is not shown in the full response in fig.9. Is the right interpretation that higher wavenumbers are cancelling out the response in the North Atlantic?

> Thank you. Indeed, the full response in (original) Fig. 9 in the North Atlantic shows positive anomalies, but the magnitude is small. On the other hand, negative anomalies over the North Pacific are large in the full response. Wave number decomposition makes wave-1 amplitude large. However, we consider the full response as more representative here, and wave-1 response is somewhat artificial, though the wave number decomposition is critical in evaluating wave propagation into the stratosphere.

8. L218-9 This could do with being more quantitative e.g. compare the simulated and observed wave-1 amplitude change averaged over 30-60N.

> Thank you for the suggestion.

  Change of wave-1 amplitude average over 30-60N is as follows.

     Observed E-W :    14.3 m
     CONV1-CNTL:    12.0 m
     CONV2-CNTL:    10.7 m

The simulated values are similar to the observed value.
We made the amplitude plot which is quite informative. We add the figure and the following text in the revised version.

"Wave amplitudes at 250 hPa for all simulations are shown in Fig. 15. Compared with the observed QBO difference (seen as EQBO minus WQBO in Fig. 9), simulated differences between CONV1 (CONV2) and CNTL are similar in magnitude and latitudinal profile. For example, simulated wave-1 amplitude averaged over 30-60°N is 12.0 m (CONV1 minus CTRL) and 10.7 m (CONV2 minus CTRL), which is in good agreement with the observed difference of 14.3 m between EQBO and WQBO. Wave-1 amplitude is also peaked at around 55°N for all cases. We also confirm that convection over the tropical western Pacific is most significant for enhanced extratropical planetary wave."

[Figure]

**Figure RC1.5. (new Fig. 15)** Latitudinal profile of the wave amplitudes at 250 hPa in AGCM simulations for November, based on 60-year mean. Black lines show the control simulation, orange lines for CONV1 simulation, and red lines for CONV2 simulation. Solid and dashed lines denote wave-1 and wave-2 components, respectively. Y-axis denotes amplitude in m.

9. L223-6 The text could do with being clearer about which results are from observations/reanalysis. (This goes for other parts of the results section as well.)

> Thank you. "Blue lines are based on reanalysis data and red lines from the simulated results."
    We added the above sentence.
    Below sentence was added for Fig. 9.
    "Red lines are based on EQBO composites and blue lines from WQBO composites."

10. Appendix B – is there a reference supporting this method?

> We are not aware of specific references. The method is a crude approximation. But, we think the horizontal advection is negligible and the balance between diabatic heating and vertical motion is kept in the tropical troposphere on a monthly time scale.

---

## Author Comment (AC3) · 6 Mar 2020

**Reply to anonymous Referee #2**

This paper presents the equatorial QBO influences on the Northern Hemisphere winter circulation. Many previous studies focused on the stratospheric pathways of this influences, while this manuscript proposes a possible mechanism for tropospheric pathways of this influences through the modulation of Rossby wave activities induced by the QBO-related convection over the tropical western Pacific and the Indian Ocean.
This topic is interesting and valuable for this scientific area. However, there are some issues as mentioned below. For these reasons, I recommend minor revisions.

We appreciate Reviewer #2 very much for the constructive comments and suggestions, in particular for detailed comments. We have carefully incorporated comments and suggestions, which, we believe, improved the manuscript in its content and presentation. Our responses to the specific comments can be found below in black (Reviewer #2's comments and suggestions) and blue (our responses).

Minor comments:

(1) p.2, l.75: The definition of QBO is based on the absolute values of equatorial zonal wind >3m/s. Please check another threshold and mention it.

>Thank you for your comment.
We tried the case with the critical zonal wind speed set to 0 m/s, which gave 14 E-QBO and 23 W-QBO winters. The OLR difference is similar to the original Fig. 3a, especially one over the tropical western Pacific is robust (Fig. RC2.1 shown below).

[Figure]

**Fig. RC2.1.** OND mean OLR difference between EQBO and WQBO winters. The criterion wind speed for QBO is 0 m/s.

We also examined the case in which the QBO was defined at 40 hPa with the 3m/s criterion, giving 18 EQBO and 14 WQBO winters. The number of EQBO winters is larger than that of WQBO winters. The OLR composite is also shown below (Fig. RC2.2). In both cases the results are not sensitive even when we change the QBO criterion slightly.

[Figure]

**Fig. RC2.2.** OND mean OLR difference between EQBO and WQBO winters. The reference height for the QBO definition is chosen at 40 hPa.

We added the following sentences at p.2, line 83.

"We also examined two cases in which we changed the threshold wind speed set to 0 m/s (14 EQBO and 23 WQBO winters) and the reference height to 40 hPa (18 EQBO and 14 WQBO winters). In both cases, the results show a high degree of robustness."

(2) p.4, l.150: The difference between Fig. 5a and 5b indicates the influence of ENSO on the equatorial east Pacific as the downward around 150W with positive OLR in Fig. 3a. Is this interference from ENSO really ruled out in later analysis?

>We think that in the present context of the QBO impacts the resemblance between composite differences (EQBO minus WQBO) with and without ENSO (both El Nino and La Nina) events mostly ruled out possible compound influences in mid- to high-latitudes from ENSO (please see Figure 7). As you pointed out there are, however, some differences in the Walker circulation between two composite differences with and without ENSO events, especially in sinking branches (Figure 5a and b). Noting this we have made a series of AGCM experiments. In addition to CONV1 (heating in the western tropical Pacific) and CONV2 (heating in the western tropical Pacific and cooling in the tropical Indian Ocean), results from the experiments with adding negative convective heating placed in the central tropical Pacific around 150°W, 0°N (CONV3P) and in the tropical Atlantic around 30°W, 10°N (CONV3A) are analyzed.

In fact, the setting for CONV3A with two sinking branches, one in the Indian Ocean and the other in the Atlantic Ocean mimics the QBO signal without ENSO most (Fig. 5b). The mid- to high-latitudes horizontal pattern in geopotential height anomalies at 250 hPa and zonal-mean zonal wind anomalies (Fig. RC2.3) are similar to the observed QBO signal (Fig. 7b). But most significantly, those horizontal and meridional patterns are captured in all experiments including CONV1 with heating only in the western tropical Pacific. We interpret this that the western tropical Pacific is the most influential to extra-tropics and polar vortex.

Above sentences are added in the revised version.

[Figure]

**Fig. RC2.3.** AGCM simulated responses of Z250 and [U]. Same as Fig. 12 but for CONV3P and CONV3A experiments.

(3) p.5, l.l.165-170: Some references are needed for the constructive interference between the anomalous Rossby wave response and the background climatological stationary wave. Smith et al. (2010) showed the linear interference between these waves in their model. Using reanalysis data, Garfinkel et al. (2010) showed the constructive interference between the ENSO-related anomaly and climatology, and Yamashita et al. (2015) showed this interference between the QBO/solar-related anomaly and the climatology.
The constructive interference in Smith et al. (2010) is linear process, thus, it is reasonable that the constructive interference is reproduced with the LBM.

Garfinkel, C. I., D. L. Hartmann, and F. Sassi (2010), Tropospheric precursors of anomalous Northern Hemisphere stratospheric polar vortices, J. Climate, 23, 3282-3299.
Smith, K. L., C. G. Fletcher, and P. J. Kushner (2010), The role of linear interference in the annular mode response to . extratropical surface forcing, J. Climate, 23, 6036-6050.
Yamashita, Y., H. Akiyoshi, T. G. Shepherd, and M. Takahashi (2015), The combined influences of westerly phase of the quasibiennial oscillation and 11-year solar maximum conditions on the Northern Hemisphere extratropical winter circulation, J. Meteor. Soc. Japan, 93, 629-644.

>Thank you for introducing appropriate references.
We added the following sentence and references.
    "The linear interference between the Rossby wave response and background climatological stationary wave has been studied in previous studies, e.g. the interference between extratropical surface forcing and the annular mode (Smith et al., 2010), the tropospheric precursor and the stratospheric polar vortex (Garfinkel et al., 2010), and the solar maximum and westerly QBO (Yamashita et al., 2015)."

(4) Some modifications of introduction are needed as mentioned below.
p.1, l.35: Holton and Tan, 1980, 1982 only show a plausible mechanism, as the latitudinal position of the zero-wind critical surface of stationary Rossby wave is primally controlled by the equatorial QBO. Recently, Watson and Gray (2014) posted this line of discussion with their model.

Watson, P.A.G., and L.J. Gray (2014) How does the quasi-biennial oscillation affect the stratospheric polar vortex?, J. Atmos. Sci., 71, 391-409

> Thank you. We modified the sentence following your suggestion.

"Holton and Tan (1980, 1982) only showed a plausible mechanism, as the latitudinal position of the zero-wind critical surface of stationary Rossby wave is primarily controlled by the equatorial QBO. Recently, Watson and Gray (2014) posted this line of discussion with their model."

p.1, l.35: "this critical latitude mechanism is not effective": The wave propagation change between the EQBO and WQBO is similar to the previous studies in highlatitudes and around equator in Naoe and Shibata (2010)'s results, in agreement with Holton-Tan relationship. In contrast, another propagation change, which is opposite to the critical line control, is analyzed in mid-latitudes by Naoe and Shibata (2010). White et al. (2015) suggested the enhanced upward wave propagation at midlatitudes due to the enhanced wave growth rather critical latitude mechanism, explaining the QBOrelated change in mid-latitudes as well as the polar vortex change in high-latitudes.

White, I.P., H. Lu, N.J. Mitchell, and T. Phillips (2015), Dynamical response to the QBO
in the northern winter stratosphere: Signatures in wave forcing and eddy fluxes of
potential vorticity. J.Atmos. Sci., 72, 4487-4507.

> Thank you. We modified the sentence following your suggestion.

"Naoe and Shibata (2010) analyzed Holton-Tan relationship by a QBO-producing chemistry-climate model (CCM) and reanalysis data. They showed the conventional critical latitude mechanism that the equatorial winds in the lower stratosphere acted as a waveguide for planetary wave propagation did not hold. White et al. (2015) suggested the enhanced upward wave propagation at mid-latitudes due to the enhanced wave growth rather than the critical latitude mechanism, explaining the QBO-related change in mid-latitudes as well as the polar vortex change in high-latitudes."

p.1, l.35: "The secondary circulation associated with the QBO in the subtropics": Naoe and Shibata (2010) and Yamashita et al. (2011) suggested the significance of the secondary circulation induced by the equatorial QBO in middle stratosphere rather lower stratosphere. In contrast, Garfinkel et al. (2012) and Lu et al. (2014) suggested the significance of the QBO-induced meridional circulation anomalies extend from the subtropics to the midlatitudes in relation to the midlatitudes change of Rossby waves due to the changes in index of refraction.

> Thank you. We modified the sentence following your suggestion.

"Naoe and Shibata (2010) and Yamashita et al. (2011) suggested the importance of the secondary circulation induced by the equatorial QBO in the middle stratosphere rather than the lower stratosphere. Garfinkel et al. (2012) and Lu et al. (2014) pointed the significance of the QBO-induced meridional circulation anomalies extending from the subtropics to mid-latitudes through changes in the refraction index and modulation of Rossby wave propagation."

Other comments:

p.4, l.155: The middle tropospheric values of red lines (WQBO) are positive and the blue lines (EQBO) are negative in Fig. 6. Does it indicate the relatively large diabatic heating in the WQBO?

> Thank you very much. The figure caption had an error. Blue lines show WQBO and Red lines show EQBO. We corrected it.

p.6, l.210: Fig. 9a shows the dipole pattern between mid-latitudes and Polar region, while Fig. 12a shows the tri-pole pattern.

> Yes, indeed. The linear response to a tropical heating is warming of the tropical troposphere. This results in positive geopotential height anomalies in the tropics and increased subtropical westerlies. We do not know how the tri-pole pattern arises, but suspect non-linear effects.

p.5, l.200: I suppose that "no interaction between the anomalous response and climatological fields" in terms of nonlinear processes, since the LBM model has the constructive interference for linear processes only.

> Thank you. We added "anomalous" and "since the LBM has the interference for linear processes only" in that sentence.

p.6, l.215: I suppose that the constructive interference is valid, when the anomalous waves and climatological waves are in phase, as the description of wavenumber 1 field at p.5, l.170. But, their wavenumber 2 fields in Fig.8 are out of phase.

>For wavenumber 2 in Figs. 8b and d, anomalies (E-W, shade) lie east of climatological trough and ridge. We made amplitude plots as a function of latitude, following Reviewer 2's suggestion.

Latitudinal profiles of wave-1 and wave-2 amplitudes for EQBO and WQBO are shown below (Figure RC1.2). "Peak values of the wave amplitude increase in EQBO Novembers both for wave-1 and wave-2 and regardless of all or non-ENSO composites." The figure and above sentence are added in the revised version. Also description for wave-2 was modified.

[Figure]

**Figure RC2.4. (new Fig. 9)** Wave amplitudes at 250 hPa as a function of latitude in the different QBO phases for November. Red (blue) solid line denotes wave-1 in the EQBO (WQBO) composite. Red (blue) dashed line denotes wave-2 in the EQBO (WQBO) composite. Y-axis denotes amplitude in m. (a) All composite. (b) Composite without ENSO winters.

Typos: p.1, l.35: atmospheric general circulation models (AGCMs)

> Thank you very much. We corrected it.

p.4, l.145: Fig. 4c, 5c -> Fig. 3c, 4c

> Thank you very much. We corrected it.